# Insights and Lessons from 3D Geological and Geophysical Modeling of Mineralized Terranes in Tasmania

**Daniel Bombardieri [1,\*], Mark Duffett [1,\*], Andrew McNeill [1], Matthew Cracknell [2] and Anya Reading [2]**

1   Department of State Growth, Mineral Resources Tasmania, Rosny Park 7018, Australia;
    andrew.McNeill@stategrowth.tas.gov.au
2   Centre for Ore Deposits and Earth Sciences (CODES), School of Natural Sciences (Earth Sciences), University
    of Tasmania, Private Bag 79, Hobart 7001, Australia; m.j.cracknell@utas.edu.au (M.C.);
    anya.reading@utas.edu.au (A.R.)
\*   Correspondence: daniel.bombardieri@stategrowth.tas.gov.au (D.B.); mark.duffett@stategrowth.tas.gov.au (M.D.)

**Abstract:** Over the last two decades, Mineral Resources Tasmania has been developing regional 3D geological and geophysical models for prospective terranes at a range of scales and extents as part of its suite of precompetitive geoscience products. These have evolved in conjunction with developments in 3D modeling technology over that time. Commencing with a jurisdiction-wide 3D model in 2002, subsequent modeling projects have explored a range of approaches to the development of 3D models as a vehicle for the better synthesis and understanding of controls on ore-forming processes and prospectivity. These models are built on high-quality potential field data sets. Assignment of bulk properties derived from previous well-constrained geophysical modeling and an extensive rock property database has enabled the identification of anomalous features that have been targeted for follow-up mineral exploration. An aspect of this effort has been the generation of uncertainty estimates for model features. Our experience is that this process can be hindered by models that are too large or too detailed to be interrogated easily, especially when modeling techniques do not readily permit significant geometric changes. The most effective 3D modeling workflow for insights into mineral exploration is that which facilitates the rapid hypothesis testing of a wide range of scenarios whilst satisfying the constraints of observed data.

**Keywords:** 3D modeling; potential field; gravity; magnetics; inversion

## 1. Introduction

In recent years, there has been widespread recognition that improving economic mineral discovery rates requires better identification and mapping of mineral systems [1–3]. As these invariably entail the movement of materials (usually fluids) within three-dimensional geological frameworks, the importance of mapping geology in 3D has become increasingly apparent. Making this mapping as accurate as possible requires the integration of geophysical information with structural interpretation from geological observation and mapping.

Over the last 20 years, there have been significant advances in 3D geological and geophysical modeling capability [4–7]. In this paper, we discuss the outcomes of 3D geological and geophysical modeling in Tasmania, starting with the Statewide 3D geological model released by Mineral Resources Tasmania (MRT) in 2002. This model was the first such construction for an entire jurisdiction in the world. Since the creation of the Statewide 3D model, there have been considerable advances in computational capacity and 3D geological and geophysical modeling software [4–7]. These enable the improved use of the 3D information inherent in gravity and magnetic field observations [8]. Rigorous integration of these data sets with each other in conjunction with geological information potentially reduces their ambiguity to a much more tractable range of plausible models [9]. However,



uncertainties remain and generally increase with distance from direct geological observations [10]. Recent computational advances facilitate methods to address this issue [7,11–14]. These advances allow us to quantify uncertainty (to a degree) of the geometry of geological units at depth through statistically generated 3D sensitivity metrics from multiple geological models that satisfy the geophysical observations [10].

We detail and discuss our modeling efforts over the last decade, and present new results using statistically generated 3D sensitivity metrics to address uncertainty. These high-resolution regional 3D models have targeted Tasmania's most important mineral provinces. Following a brief review of the geological context, the models are presented via grouping into Western and Eastern Tasmanian terranes. Figure 1 illustrates the Western Terrane model localities, which include the Rosebery Region 3D model and its derivatives, (i.e., Rosebery–Lyell, Rosebery–Pieman 3D models). These models have stimulated both exploration (e.g., gravity surveys and drilling) and research efforts in the form of Honors and PhD projects, (e.g., [15]). Eastern Terrane models (Figure 2) were constructed to cover the major orogenic gold and zoned Zn, Sn, and Cu deposits of Northeast Tasmania. These include the Lebrina, Alberton-Mathinna, and Scamander 3D models.

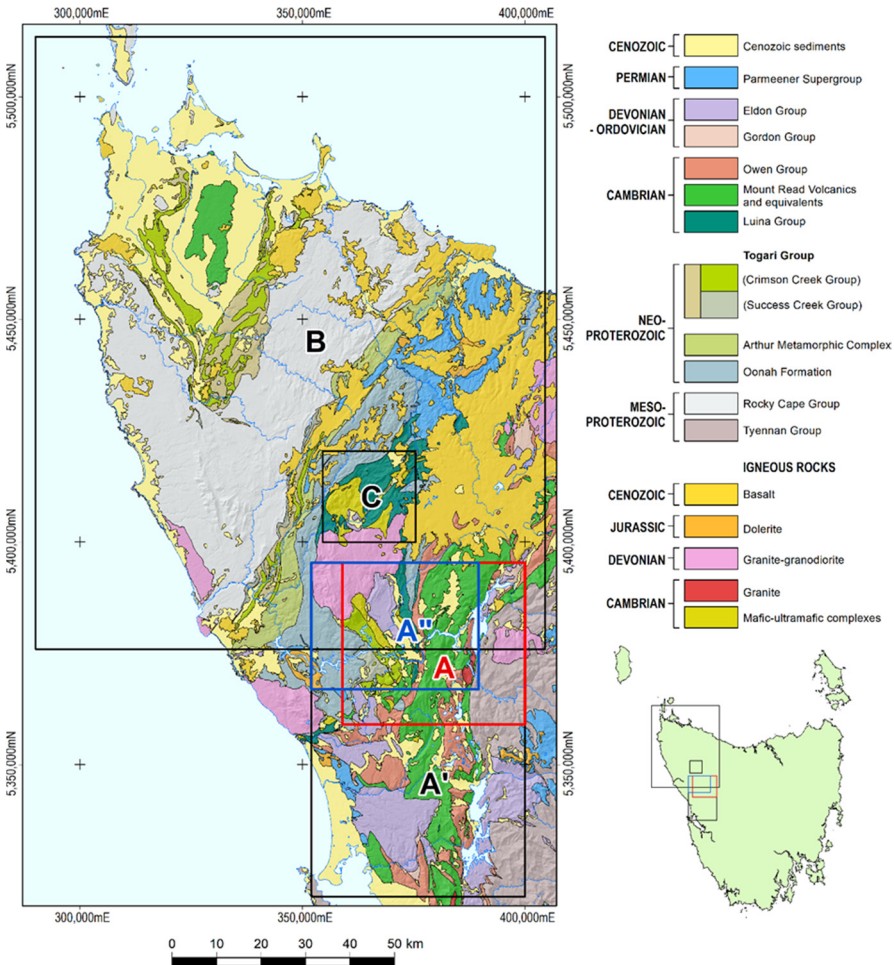

**Figure 1.** Geology of the Western Tasmanian Terrane and associated locality map. A (red rectangle), A' (black rectangle), and A'' (blue rectangle) represent the Rosebery Region, Rosebery–Lyell, and Rosebery–Pieman 3D model extents. B and C represent the Northwest Tasmania and Heazlewood–Luina–Waratah 3D model extents. Coordinates in this and subsequent figures are of the Map Grid of Australia, zone 55.

Finally, we discuss the insights and lessons from these modeling efforts and briefly discuss avenues for improvement.

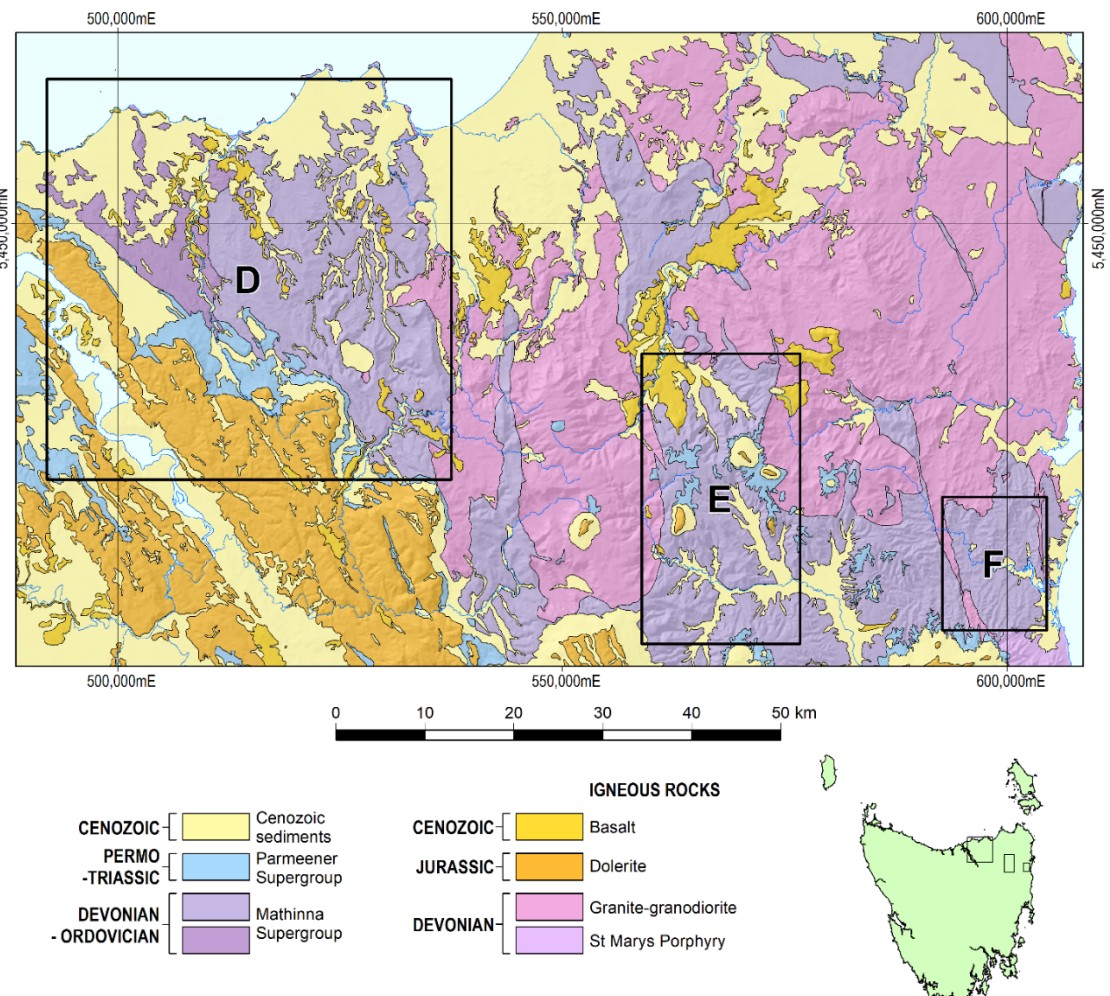

**Figure 2.** Geology of the Eastern Tasmanian Terrane and associated locality map. D, E, and F represent the Lebrina, Alberton–Mathinna, and Scamander 3D model extents, respectively.

## 2. Tectonic and Geological Setting

Tectonically, Tasmania can be divided into Western Tasmanian and Eastern Tasmanian Terranes [16]. The Eastern Tasmanian Terrane (ETT) is characterized by Paleozoic turbidites and granitoids. In contrast, the Western Tasmanian Terrane (WTT) has a longer tectonic history and comprises Mesoproterozoic to Paleozoic strata and igneous lithologies [17]. The two terranes were juxtaposed during the middle Devonian Tabberabberan Orogeny [18] and both terranes are variably covered by Carboniferous to recent sedimentary and igneous rocks.

### 2.1. Western Tasmanian Terrane

The geology and tectonic structures of Western Tasmania are complex due to a series of major tectonic events and orogenies. The oldest lithologies include the Mesoproterozoic Rocky Cape Group and correlates, which comprise a ~10 km thick siliciclastic shelf sequence [18–20]. The Rocky Cape Group is generally mildly deformed and metamorphosed to greenschist facies.

Neoproterozoic rift-facies sediments and volcanic sequences such as the Oonah Formation and correlates, Success Creek, and Crimson Creek Formations preceded the Tyennan Orogeny, which resulted in the west-directed emplacement (over-thrusting) of large allochthonous sheets of mafic and ultramafic rocks and mélanges comprising mudstone, chert, and basalt (Luina Group) [21–24]. Relicts of the allochthonous sheets include the Heazlewood River, Wilson River, and Huskisson River ultramafic complexes. The Tyennan

Orogeny also led to eclogite facies metamorphism of the structurally complex Tyennan Region occupying much of Central Tasmania [25] (p. 16) and the Arthur Metamorphic Complex, a NNE-trending high-strain belt less than 10 km wide extending across Northwest Tasmania [26].

Middle Cambrian post-collisional magmatism produced the mineral-rich Mount Read Volcanics (MRV) belt extending through Western and Northwestern Tasmania. The MRV have a highly complex internal architecture comprising felsic and mafic lavas, syn-volcanic intrusions, and syn-eruptive volcaniclastic phases [27]. The MRV were accompanied by the emplacement of sub-volcanic Cambrian granites, including the Murchison and Darwin Granites [17,28,29].

The MRV are unconformably overlain by the siliceous Owen Conglomerate, which also unconformably overlies the older Cambrian and Proterozoic basement [30]. The Owen conglomerate is overlain by the dominantly shallow marine carbonates of the Gordon Group (Ordovician to lower Silurian) and the siliciclastic, with minor limestone, shelf sequence of the Silurian to lower Devonian Eldon Group [19].

The Devonian Tabberabberan Orogeny caused regional scale folding, reactivation of the Cambrian basinal fault architecture, and the intrusion of a considerable volume of associated post-tectonic granites [24,31,32].

### 2.2. Eastern Tasmanian Terrane

The Mathinna Supergroup, an Ordovician to Early Devonian turbiditic sequence with cumulative stratigraphic thicknesses exceeding 10 km, is the oldest unit exposed in the Eastern Tasmanian terrane (Figure 2), although Cambrian ultramafic rocks (with different geochemical affinities to the ultramafics associated with the Western Terrane) are interpreted on geophysical evidence to underlie much of the northern half of the Eastern Tasmania Terrane (ETT) [33]. Following two major deformational episodes during the Tabberabberan Orogeny, in the course of which Western and Eastern Tasmania reached their current juxtaposition [19,32,34–36], the Mathinna Supergroup was intruded by granitoids of Devonian age that include the Scottsdale Batholith (incorporating the Tombstone Creek and Russells Road Granites) and Blue Tier Batholith (incorporating the Pyengana Granodiorite, Poimena, and Mt. Paris Granites). Many of these plutons are characterized as I-type, having been derived from the partial melting of igneous source rocks [32].

### 2.3. Younger Rock Sequences

The basement rocks in both the Western and Eastern Terranes are unconformably overlain by Carboniferous to Triassic sediments of the Parmeener Supergroup, which have been extensively intruded by sheets of Jurassic dolerite typically hundreds of meters thick. In the late Cretaceous onshore and offshore basin, development commenced as a result of rifting from Antarctica and spreading in the Tasman Sea. These basins have filled with local basement-derived sediments and are associated with more widespread Cretaceous to Miocene basaltic lava flows and local volcanic centers [37,38].

Apart from the presence of locally developed placer deposits, including tin, gold, and platinum group elements, and bauxite developed on weathered dolerite and basalt, these younger rocks are not significant targets for metallic mineral exploration.

### 2.4. Economic Geology

Tasmania has a remarkably rich and diverse mineral endowment proportional to its size. Polymetallic Cambrian volcanogenic-hosted massive sulfide (VHMS) deposits hosted by the Mount Read Volcanics have supported profitable mining operations for well over a century, with major examples including Rosebery (Zn-Pb-Ag-Au) and Mount Lyell (Cu-Au). A second major metallogenic episode arose from widespread Devonian granitoid intrusion, resulting in major tin ore deposits in the WTT, at Mt Bischoff and Renison Bell (both of which have been the largest hard-rock Sn mines in the world at different times) and extensively in the ETT, both hard rock and alluvial. Orogenic gold is a third class of

mineralization prevalent in Tasmania, particularly the ETT, with over a million ounces extracted thus far [28].

## 3. Inversion Methodology and Data

The method of fully constrained lithological cooperative inversion is successfully achieved by first creating a starting geological model (henceforth called the "reference model") that is consistent with all the geological observations, and closely honors the observed geophysical response. This is necessary because the Monte Carlo techniques used in sensitivity modeling adopt very detailed processes for changing individual properties or geometries. In addition, a close starting model also ensures that the time needed to explore probability space is minimized and the inversion converges to an acceptable misfit [4,6,7,10].

### 3.1. Reference Model Geophysical Validation

Using the methodology described above and illustrated in Figure 3, we first construct the "reference model" in GoCAD^TM [5,10]. The model comprises geological interfaces (i.e., stratigraphic horizons and faults) that honor the observed data (e.g., geological maps, drillholes, and cross sections; see Figure 4A). The volumes thus defined are discretized (Figure 4B) in preparation for forward and inverse modeling using GoCAD's potential field module (VPmg code; [5]). VPmg solves the perturbation equations using the Method of Steepest Descent, resulting in rapid inversion speed. During inversion, the degree of fit is judged according to the magnitude of the chi-squared misfit. Inversion stalls or finishes, if it reaches the maximum allowed iterations or further iterations, do not reduce the misfit [5].

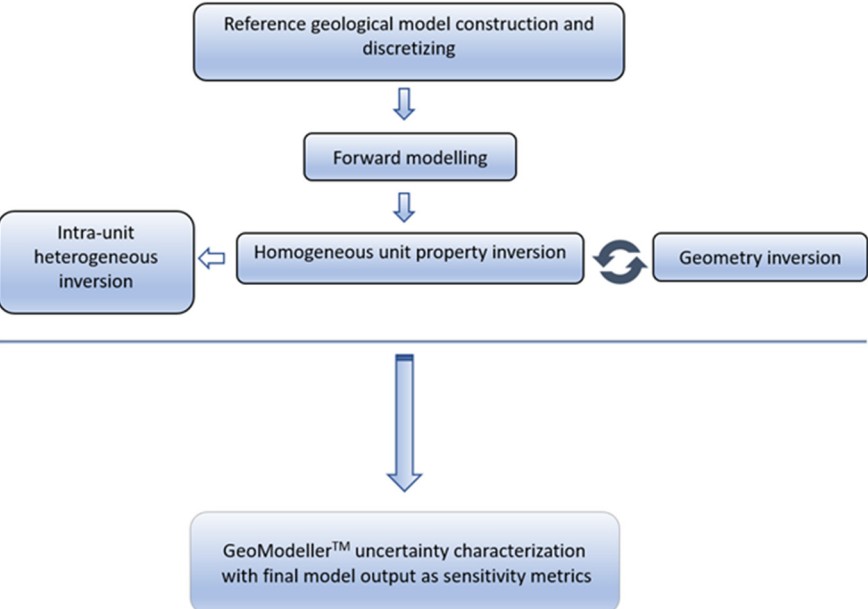

**Figure 3.** Flow chart of the applied methodology for 3D model generation and sensitivity characterization. Intra-unit heterogeneous inversion (GoCAD^TM) was used specifically for the Rosebery–Lyell 3D model to generate rock property statistics on mafic–ultramafic complexes, which were used to inform follow-up modeling efforts.

Homogeneous (single property value for an entire unit) and limited geometric inversion is undertaken to ensure that unit properties correctly represent their bulk response and to ensure reasonable input for the next stage of modeling. However, the 3D model derived to this point is a "best estimate" synthesis only and as is well-known with potential fields, this solution is not unique [10].

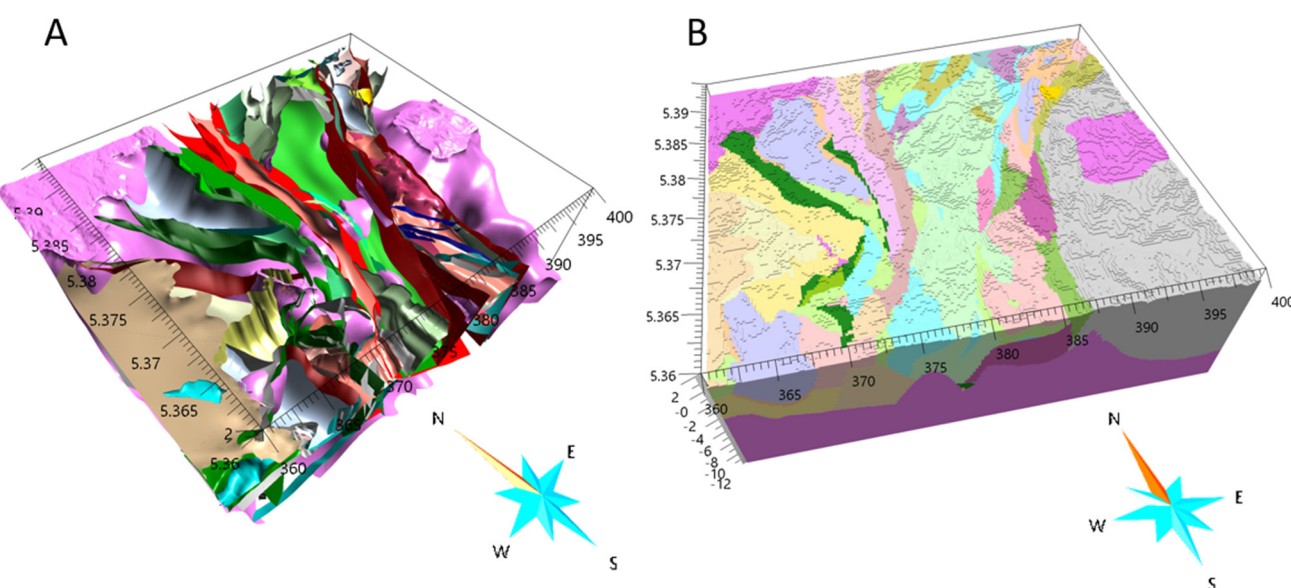

**Figure 4.** (**A**) 3D schematic of the Rosebery region illustrating the geological model lithologies. Notable features include granitoids (pink surfaces) and complex fault network architecture. The x- and y-axes are measured in meters east ($\times 10^3$) and north ($\times 10^6$), respectively. The z-axis is in kms. These axes measurements apply to subsequent models. (**B**) Discretized voxet reference model of the Rosebery region.

The GoCAD$^{TM}$ voxet reference model (which closely honors the observed geophysical response) is imported into GeoModeller$^{TM}$ for cooperative gravity and magnetic inversion and sensitivity characterization.

### 3.2. Cooperative Inversion and Sensitivity Modeling

Addressing the ambiguity inherent in potential field modeling, GeoModeller$^{TM}$ was employed to both refine the inversion and explore the range of similarly plausible possible models, with the goal of estimating the spatial variability of confidence in the model elements [10].

The stochastic exploration algorithm used takes a Bayesian approach, Markov chain Monte Carlo simulation, generating a sequence of linked models starting with the initial model making small "random" changes to the lithological boundaries and physical properties. The proposed change is then accepted or rejected, depending first on whether geometric constraint parameters designed to enforce geological reasonableness ("Commonality" and "Shape Ratio") are satisfied, and second on whether the geophysical response misfit is improved or maintained below an acceptable threshold [10]. Physical properties within modeled units are constrained to a preset distribution estimated from the petrophysical database.

Cooperative gravity and magnetic inversion allow volume and depth information to be determined. However, before this occurs, a geological test takes place prior to a property (geophysical) test. This geophysical test only takes place if the preceding geological test (boundary change) is accepted; in other words, a geological boundary change acceptance triggers a property change that is then tested [10].

Upon completion of the inversion run, GeoModeller carries out a statistical analysis of the ensemble of stored models that reproduced the observations to an acceptable degree [10]. Sensitivity metrics include the mean density and susceptibility derived from the accumulated accepted inversion proposals/models. The probability threshold which records the lithology assigned to that voxel in at least 99% of the acceptable models, and the most probable model which records the lithology most often assigned to a model voxel location from the ensemble of acceptable models [10].

### 3.3. Potential Field Data Preparation—Regional-Residual Separation

The potential field contribution from any magnetic/gravity source located beyond the extents (laterally and vertically) of the local model can interfere with this model and be erroneously attributed to its response when the data are inverted. To account for this regional response, a coarse (1 km × 1 km × 250 m cell size) regional model approximately three times the extent of the local model was constructed, and an unconstrained inversion completed on this volume. The resulting property distribution reproduces the regional gravity and magnetic fields arising from sources outside the volume of interest to an acceptable degree. The GoCAD^TM software allows for the local model to be incised into the regional model as part of the inversion workflow, and the regional trends are thus accounted for in the forward and inverse model computations. Subsequently, to use the sensitivity modeling environment provided by GeoModeller^TM, regional gradient and edge effects were addressed by detrending and padding [10].

Following unsatisfactory early inversion attempts, an additional step of upward continuation by 200 meters was applied to both gravity and magnetic data used in modeling. This had the effect of suppressing overfitting that would otherwise produce geologically implausible high-frequency variations in inverted model features.

### 3.4. Data Sources

All the geological and geophysical modeling described herein is based fundamentally on the regional geological mapping and geophysical data acquisition conducted or compiled by MRT over many decades. All these data are publicly available from MRT. In all areas where 3D geophysical modeling has been applied, the magnetic data used have generally been flown at 200 m line spacing with ground clearance around 80 m. Gravity observations are less regular; however, station spacings around 1 km are typical in the areas that have been modeled geophysically. Complete Bouguer anomaly (CBA, i.e., terrain-corrected) values have been computed for each station [39]. As is generally the case with all gravity modeling applied to intra-crustal features in Tasmania, Bouguer anomaly values referred to throughout herein are residuals derived from a single crust–mantle–ocean model [40]. This has the effect of removing gradients arising from the continental edge and associated Moho topography that would otherwise dominate the observed field.

### 3.5. Rock Physical Properties

Initial unit bulk physical rock properties and allowed ranges for susceptibility and density for all Western and Eastern Terrane models are illustrated in Tables 1–4. These were sourced from values adopted from the literature [10,15,33,41] and MRT's publicly available physical rock property database, henceforth referred to collectively as "existing values".

**Table 1.** Western Terrane existing and model-informed mean density (kg/m$^3$).

| Lithology | Existing [1] | Output [2] | 1 σ |
| --- | --- | --- | --- |
| Jurassic Dolerite | >2.77 | 2.84 | 0.043 |
| Crimson Creek Formation [3] | 2.77–3.50 | 2.97 | 0.060 |
| Devonian Carboniferous granites | 2.60–2.62 | 2.60 | 0.006 |
| Eldon Group Sediments | 2.67–2.69 | 2.69 | 0.018 |
| Gordon Group | 2.67–2.72 | 2.72 | 0.013 |
| Owen Group Sediments | 2.67–2.72 | 2.72 | 0.020 |
| Tyndall Group | 2.68–2.74 | 2.69 | 0.015 |
| Western Volcano—Sedimentary Sequence. | 2.68–2.74 | 2.69 | 0.011 |
| Farrell Slates | 2.67–2.74 | 2.71 | 0.001 |
| Que Hellyer Volcanics | 2.72–2.79 | 2.79 | 0.010 |
| Cambrian Granites | 2.64–2.75 | 2.67 | 0.010 |
| Andesites | 2.71–2.82 | 2.75 | 0.020 |
| Quartz Feldspar Porphyry | 2.64–2.72 | 2.66 | 0.015 |
| Central Volcanic Complex | 2.71–2.75 | 2.71 | 0.006 |
| Eastern Quartz Phyric Seq. | 2.67–2.68 | 2.64 | 0.017 |
| Sticht Range Beds | 2.65–2.68 | 2.64 | 0.013 |
| Mafic-Ultramafic Complexes [4] | 2.75–2.97 | 2.81 | 0.047 |
| Mafic-Ultramafic Complexes [5] | 2.65–2.75 | 2.68 | 0.041 |
| Mafic-Ultramafic Complexes [6] | 2.55–2.65 | 2.62 | 0.048 |
| Luina Group | 2.78–2.82 | 2.79 | 0.018 |
| Crimson Creek Formation | 2.72–2.77 | 2.77 | 0.016 |
| Success Creek Group | >2.74 | 2.73 | 0.013 |
| Oonah Formation | 2.67–2.72 | 2.69 | 0.005 |
| Proterozoic Basement | 2.67–2.72 | 2.69 | 0.011 |
| Proterozoic Basement Magnetic [7] | 2.75–2.80 | 2.77 | 0.013 |

Note: [1] denotes existing values sourced from [10,42,43], and MRT's publicly available physical rock property database, [2] denotes GeoModeller$^{TM}$ informed sensitivity statistics for mean density from an ensemble of acceptable models, [3] denotes lithology and density for typical Mt Lyndsay skarn material sourced from MRT's publicly available physical rock property database. Superscripts [4], [5], and [6] represent a range of density values informed from GoCAD$^{TM}$ heterogeneous inversion modeling. [7] denotes density value for magnetic Proterozoic basement informed from 2D gravity modeling [44].

**Table 2.** Western Terrane existing and model-informed mean susceptibility (SI × 10$^{-3}$).

| Lithology | Existing [1] | Output [2] | 1 σ |
| --- | --- | --- | --- |
| Jurassic Dolerite | >10 | 20 | 0.001 |
| Crimson Creek Formation [3] | 150 | 100 | 0.147 |
| Devonian Carboniferous Granites | 0 | 0 | 0.003 |
| Eldon Group Sediments | 0 | 0 | 0.000 |
| Gordon Group | 0 | 0 | 0.000 |
| Owen Group Sediments | 0 | 0 | 0.000 |
| Tyndall Group | 13 | 6 | 0.003 |
| Western Volcano—Sedimentary Sequence | 2 | 1 | 0.001 |
| Farrell Slates | 1 | 1 | 0.001 |
| Que Hellyer Volcanics | 4 | 1 | 0.002 |
| Cambrian Granites | 35 | 37 | 0.001 |
| Andesites | 4 | 4 | 0.001 |
| Quartz Feldspar Porphyry | 2 | 2 | 0.001 |
| Central Volcanic Complex | 2 | 2 | 0.002 |
| Eastern Quartz Phyric Seq. | 13 | 13 | 0.002 |
| Sticht Range Beds | 0 | 0 | 0.002 |
| Mafic-Ultramafic Complexes [4] | 0–50 | 24 | 0.038 |
| Mafic-Ultramafic Complexes [5] | 50–200 | 150 | 0.028 |
| Mafic-Ultramafic Complexes [6] | 200–400 | 194 | 0.035 |
| Luina Group | 12 | 10 | 0.054 |
| Crimson Creek Formation | >65 | 11 | 0.011 |
| Success Creek Group | 0–3 | 0 | 0.001 |
| Oonah Formation | 0–6 | 1 | 0.002 |
| Proterozoic Basement | 0–2 | 0 | 0.001 |
| Proterozoic Basement Magnetic [7] | 30–35 | 40 | 0.030 |

Note: Superscript [1] denotes existing values sourced from [10,42,43] and MRT's publicly available physical rock property database, [2] denotes GeoModeller$^{TM}$-informed sensitivity statistics for mean susceptibility (SI × 10$^{-3}$) from an ensemble of acceptable models, [3] denotes lithology and susceptibility for typical Mt Lyndsay skarn material sourced from MRT's publicly available physical rock property database. Superscripts [4], [5], and [6] represent a range of susceptibility values informed from GoCAD$^{TM}$ heterogeneous inversion modeling. [7] denotes susceptibility value for magnetic Proterozoic basement material as modeled by [58].

**Table 3.** Eastern Terrane existing and model-informed mean density (kg/m$^3$).

| Lithology | Existing [1] | Output [2] | 1 σ |
|---|---|---|---|
| Cenozic Basalt | 2.85–2.90 | 2.84 | 0.100 |
| Jurassic Dolerite | 2.80–2.90 | 2.79 | 0.020 |
| Parmeener Supergroup | 2.40–2.55 | 2.53 | 0.004 |
| Devonian Carboniferous Granites | 2.60–2.62 | 2.62 | 0.010 |
| Granodiorites | 2.70–2.75 | 2.70 | 0.001 |
| Magnetic Granodiorites | 2.71 | 2.74 | 0.010 |
| Mathinna Supergroup [3] | 2.73–2.76 | 2.75 | 0.010 |
| Scamander Formation [4] | | 2.73 | 0.010 |
| Lone Star Siltstone [5] | | 2.74 | 0.010 |
| Cambrian Sediments | 2.65–2.79 | 2.74 | 0.010 |
| Mafic–Ultramafic Complexes | 2.55–2.97 | 2.65 | 0.010 |
| Proterozoic Basement | 2.67–2.72 | 2.69 | 0.010 |

Note: [1] denotes existing values sourced from [10,33] and MRT's publicly available physical rock property database, [2] denotes GeoModeller$^{TM}$—informed sensitivity statistics for mean density (kg/m$^3$) from an ensemble of acceptable models, [3] denotes density value range for Mathinna Supergroup sourced from MRT's publicly available physical rock property database. [4,5] denotes model informed individual unit density for Mathinna Supergroup subunits.

## 4. Statewide Model

### 4.1. Construction

The 3D statewide geological model [45] (p. 14) released by MRT in 2002 was the first such construction for an entire jurisdiction in the world (Figure 5). It was built from 51 interpretive cross-sections across the entire state, mostly at 10 km spacing and E-W orientation, interpreted to an average depth of 7 km. Sections were completed within six tectonic elements [45] on a domain by domain basis. Aspects of each element in terms of rock associations, tectonic setting and structure, intrusions, and mineralization are discussed in [42]. Additional cross-sections showing greater detail and complexity were developed for the mineral-rich central MRV belt (15 × 5 km-spaced E-W sections and 2 N-S sections).

Interpretation of features at depth on the cross-sections was strongly informed by automated potential field source edge mapping ("worms"; [46]) and 2D forward models [45]). Stacked worm profiles derived from various continuation heights (thereby related to source depth; [46,47]) and attributed by amplitude were used extensively for determining the relative shape, depth extent, and continuity of edges (e.g., faults, intrusive boundaries). Additional geological information included the Pasminco Mt Read model for Western Tasmania [45], granite isopachs derived from previous 2D gravity and magnetic modeling [48], and regional onshore and offshore reflection seismic sections [49].

Each section was interpreted according to a standard legend condensing Tasmania's geology to 28 rock units for modeling. Adjacent sections were drawn by alternating authors to reduce the potential of individual bias. Hand-drawn sections were imported into 3D space for model construction by explicit digitizing and wire framing. Several iterations were required to resolve issues that arose as the 3D picture unfolded, which had not been apparent on 2D sections considered in isolation.

### 4.2. Analysis

The 3D worm data (the depth dimension being derived from the level of upward continuation) were used to classify and weight edges represented in the model by their continuity in depth and length [45]. Deeper-penetrating structures thus identified, such as the Henty Fault, are inferred to have greater metal-bearing fluid carrying capacity. Comparison with known major VHMS mineral systems such as Rosebery, Mt Lyell, Henty, Hercules, and Hellyer showed a strong association with northwest-trending structures segmenting the MRV [45] (pp. 44–45). The northern MRV seems to hold the best potential for the repetition of similar structural positions; however, NW-trending structures are also common in the MRV south of Macquarie Harbor, presenting further exploration targets.

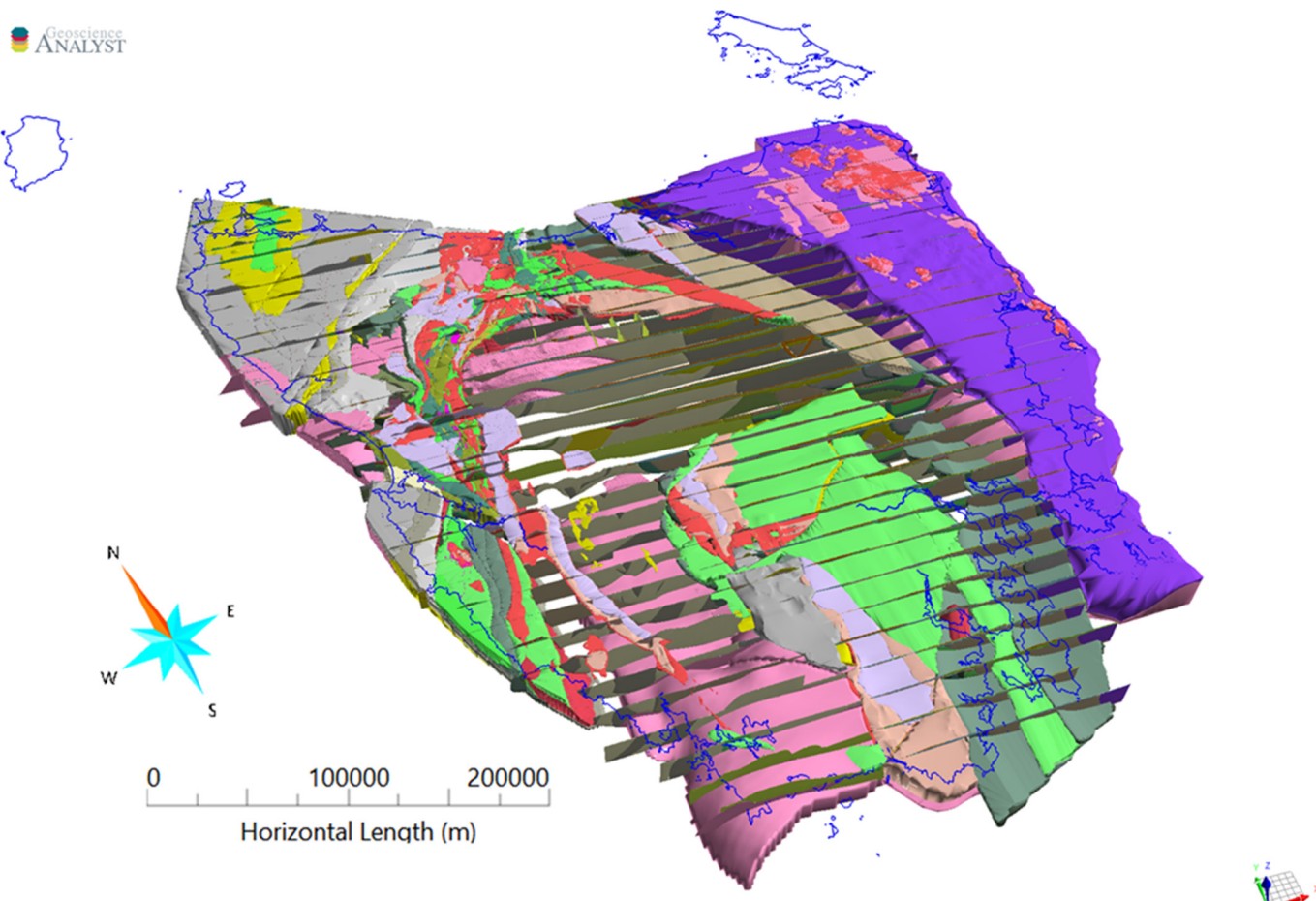

**Figure 5.** The Statewide 3D geological model of Tasmania including cross-sections. The world's first 3D model of an entire jurisdiction.

While proximity to faults is somewhat correlated with intrusion-related deposits in Western Tasmania, including Mt Bischoff, Renison Bell, and Avebury, edge detection analysis indicated that fault magnitude and orientation controls on these are not as evident as with VHMS deposits. This suggests that fault length (and by inference depth) is not as critical a factor, though fault intersections with granite remain important [45] (p. 48). However, the NW-trending Federal–Bassett fault zone on which Renison Bell is situated is a continuation of the corridor associated with VHMS deposits at Hercules and Henty, suggesting that this structure was a fluid conduit since the Cambrian, and fault intersections weighted by worm magnitude do have some association with intrusion-related mineralization. This analysis highlights similar areas with strong exploration potential [45].

Orogenic gold deposits mainly occur in the ETT. Curiously, in one of the more prolific zones, the NNE-trending "Gold Corridor", gold mineralization appears almost antithetic but sub-parallel to the total edge length. This and the other relationships observed between known Au mineralization and total edge length perhaps suggest that metal-bearing fluids migrate away from potential source structures further into the surrounding country rocks before deposition. However, some clusters of gold occurrences appear related to isolated regions of high fracture density tapping deeper-seated structures, indicated by high values of intersection-weighted edge detection [45] (pp. 52–53). Not all such regions are associated with known mineralization, but rather are indicated for follow-up exploration [45].

### 4.3. Impact

The Statewide 3D model was an ambitious undertaking given Tasmania's geological and metallogenic complexity, engaging over half a dozen geoscientists over a period exceeding 18 months. Though it is impossible to isolate and quantify contributing factors,

the release of the Statewide 3D model coincided with a doubling of exploration expenditure in Tasmania, with new ideas being generated and pursued. It became the framework underpinning subsequent, more focused modeling efforts. However, it is difficult to gauge the extent to which the Statewide 3D model has been directly applied in exploration target generation. Some anecdotal evidence suggests that application may have been retarded to some extent by significant limitations in 3D visualization platforms freely available at the time (early–mid-2000s) and the difficulty of integrating the released data package with other data sets held by individual explorers.

For the most part, the key features of this model remain valid; however, the advent of new data and geophysical interpretation techniques has brought some inconsistencies to light. Nevertheless, there have been no further editions of the statewide model, due at least in part to the difficulty, inherent to explicit modeling, of making substantial alterations to a model of its size and complexity.

## 5. Western Tasmanian Terrane Models

Commencing around 2010, MRT embarked on the development of a second generation of regional 3D models. These include the Rosebery Region, Rosebery–Lyell, and Rosebery–Pieman 3D models. As well as incorporating higher levels of stratigraphic and structural detail, a key difference with the Statewide model was the incorporation of fully three-dimensional magnetic and gravity modeling, enabled by advances in software capability. In combination with bulk geological unit physical property estimates, this provides considerably tighter constraints on model geometry.

### 5.1. Rosebery Region

The Rosebery Region 3D model (Figure 1A) spans 48 km (west to east) by 45 km (north to south), encompassing some of the most highly mineralized crust on the planet, with the Rosebery polymetallic ore mineral system and carbonate-replacement Sn deposits at Renison Bell and Mount Lindsay being foremost among many others.

The Rosebery Region model (Figure 4A) was constructed in a similar fashion to the 2001 Statewide 3D model, but at a scale that enabled the incorporation of finer details from structural and geophysical model sections constructed by previous workers [42,43,50]. Reconciliation of these sections required the exercise of geological judgement to produce plausible geometries consistent with outcrop and drillhole constraints. Stratigraphic horizon and fault surfaces were constructed individually from their curve traces on the geological map and cross-sections. Curves were linked into individual groups and assigned horizon and fault codes. An interpolation algorithm was then used to create triangulated mesh surfaces representing each fault and stratigraphic horizon.

### 5.1.1. Forward and Inverse Modeling

Forward and inverse modeling was carried out using the vertical prism-based VPmg$^{\text{TM}}$ software [5,51,52]. For the Rosebery Region and subsequent derivative models, magnetic and gravity observations were upward continued to 200 m to remove the effects of short-wavelength features too fine for practical representation at the modeling scale. In preparation for forward and inverse modeling, the geological unit geometry was discretized into $200 \times 200 \times 100$ m voxels for geophysical modeling, resulting in 6,049,296 cells.

An iterative series of magnetic/gravity forward and homogeneous inversion modeling tasks (i.e., homogeneous single property value for an entire unit) was undertaken. The homogeneous inversion fine tunes the initial bulk unit magnetic susceptibility and density estimates obtained mainly from sample measurements, as the latter may not be representative of the entire stratigraphic units. Subsequent geometry inversion (geometry modification) allows variation of geological boundaries. Small changes (~2% per iteration) were applied while always honoring designated geological constraints.

### 5.1.2. Magnetic Response

Within the Rosebery region, the magnetic response is dominated by allochthonous fault-bounded slices of serpentinized ultramafic material (Figure 6). We therefore decided to examine the geometry of these units first on the grounds that there were fewer units to deal with and that magnetic responses have a greater sensitivity than gravity to the source depth and dip. Resolution of the geometry of these units thus effectively provides a relatively firm structural framework for the remainder of the model.

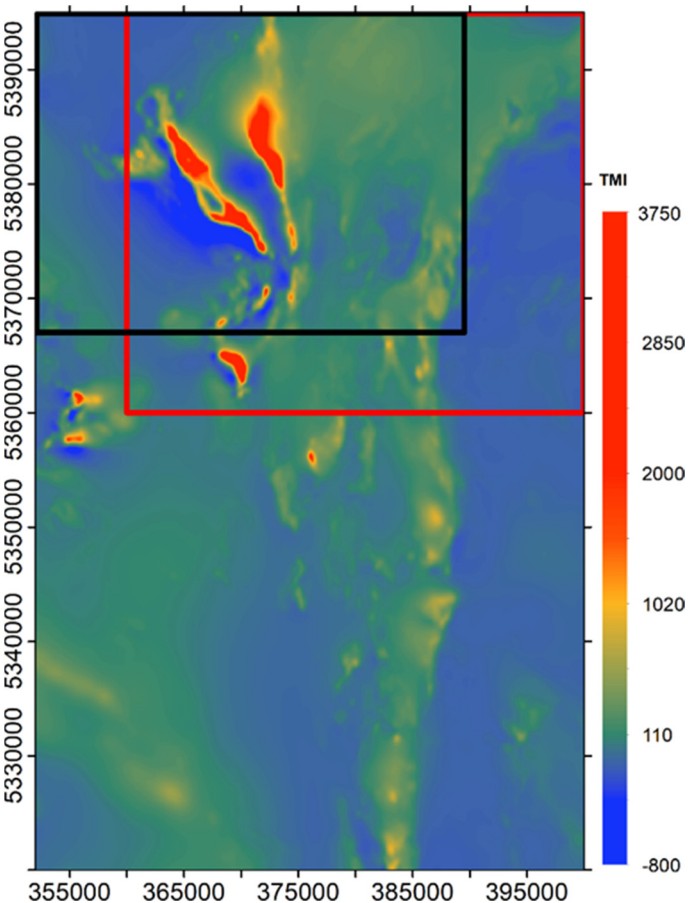

**Figure 6.** The observed total magnetic intensity (TMI, nT) highlighting the response of the various lithologies that contribute to the local magnetic field. D model extents. Features exhibiting the highest magnetic responses are associated with allochthonous fault-bounded slices of serpentinized ultramafic material ("ultramafic complexes"). The x- and y-axes are measured in meters east and north, respectively. These measurements apply to all subsequent grid images.

Initial forward and homogeneous modeling results showed a reasonable correspondence with the overall large-scale observed magnetic features including bulk unit physical rock property values. Figure 7 illustrates the magnetic residual misfit after homogeneous inversion and geometry modification. Large variations in apparent magnetization within volumes strongly inferred to consist of ultramafics (from outcrop) suggest that very large ultramafic magnetic susceptibility variations indicated by the petrophysical database are often spatially coherent. These variations were investigated in subsequent modeling efforts (see Section 5.2.1).

Remaining unresolved broad-scale misfits from the starting forward model were mostly accounted for by additional ultramafic material at depth in a configuration consistent with the regional structural style (listric thrust detachments). In contrast, finer-scale features in the magnetic residual misfit are more confidently ascribed to volumes that are magnetically anomalous in a geological as well as geophysical sense.

### 5.1.3. Gravity Response

The Bouguer gravity anomaly (Figure 8) was modeled via the same workflow as for the magnetics. The gravity residual misfit after homogeneous inversion (Figure 9) showed some long-wavelength residual features largely ascribed to inadequate model geometry. These largely account for the failure of homogeneous inversion to reach the estimated gravity data noise envelope of 0.5 mGal. Notably, several residual misfits are associated with areas of poor gravity station coverage, suggesting the possibility that the gravity data may be being inadequately interpolated between the sparse observation points. Other residual misfits with wavelengths shorter than model unit dimensions are ascribed to intra-unit density variation possibly arising from alteration or facies changes.

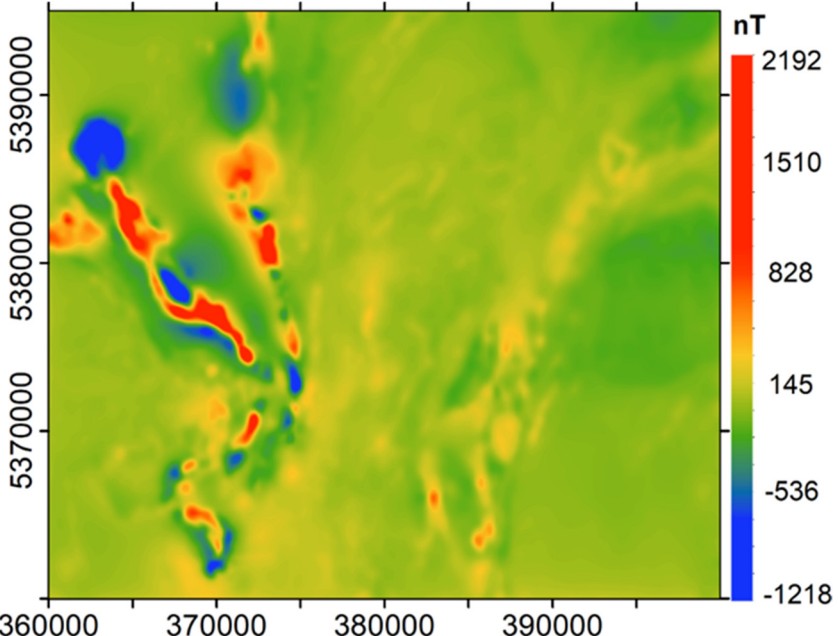

**Figure 7.** The TMI residual misfit of the Rosebery Region reference model (i.e., the difference between the observed and calculated responses; rms misfit 175.81 nT). Red residual anomalies indicate that additional magnetic material is required to reduce the misfit to match the observed TMI response. In contrast, blue residuals indicate that less magnetic material is required to reduce the misfit.

A negative residual of −10 mGal amplitude near the southwestern Huskisson Syncline (Figure 9) was informally named the Tributary Creek gravity anomaly (TCGA). Geometry inversion was then conducted within the fixed framework defined by the magnetic units (principally the mafic–ultramafic complexes in the vicinity of the TCGA). This resulted in a substantial change to the initial Devonian granite body (Figure 10), with the re-modeled intrusion rising to within ~1000 m of the surface and intersecting both ultramafics and Gordon Group carbonates. Both scenarios have potential for generating base metal mineralization, particularly nickel and tin. More broadly, the gravity-controlled geometry inversion resulted in substantially less volume of Devonian granite at depth replaced by Neo Proterozoic basement rocks. However, the Devonian granite was generally shallower (i.e., closer to the surface) and with greater relief.

The TCGA area consequently became a focus for further commercial exploration. A total of 181 new gravity stations were obtained near the TCGA [53]. These new data confirmed the likelihood of a volume of low-density material at this location, which was interpreted as granite possibly intruding mafic–ultramafic complexes and/or Gordon Group carbonates [53]. However, as the core of the TCGA was still defined by only a few gravity stations, Yunnan Tin, who held the exploration license over the area, acquired more detailed gravity data (~100 m spacing; [54]) to confirm its location and character. These new data were used to inform later modeling described below.

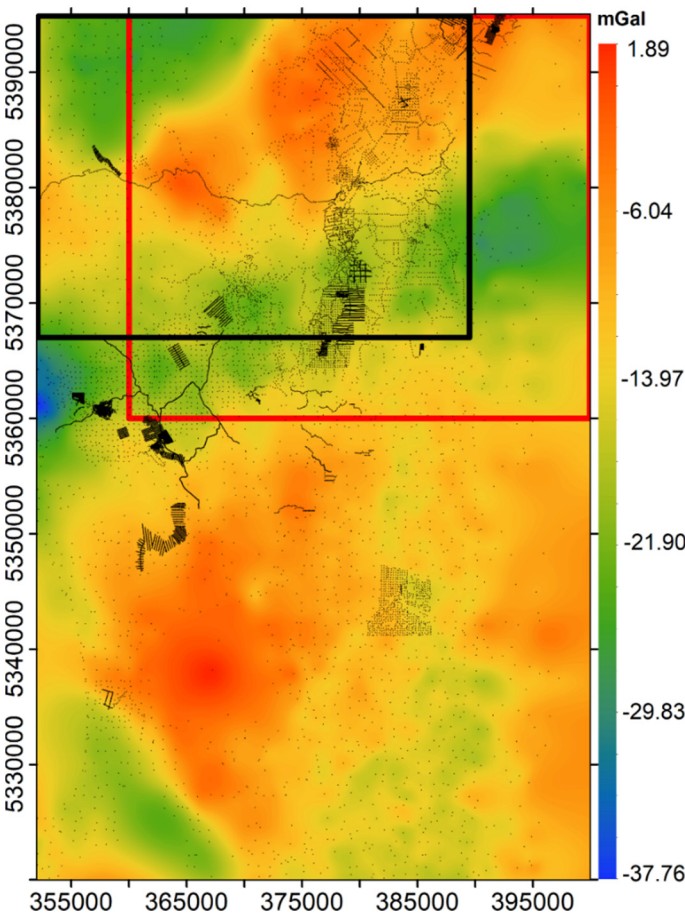

**Figure 8.** Illustrates the observed gravity response (mGal; including gravity stations, black dots) highlighting the response of the various lithologies that contribute to the local gravity field. Red and black rectangles represent the Rosebery Region and Rosebery–Pieman study area, respectively. The full grid represents the Rosebery–Lyell 3D model extents.

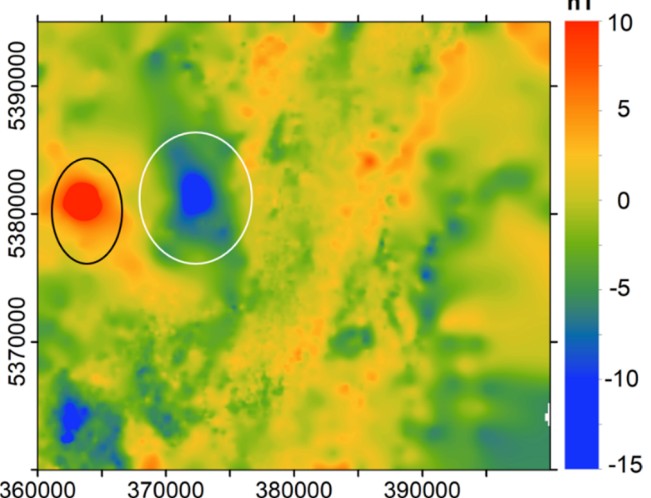

**Figure 9.** The gravity residual misfit of the Rosebery Region reference model. The rms misfit was 2.29 mGal. Note the longer-wavelength negative residual (shown by the white ellipse) and positive residual (shown by the black ellipse) associated with the Huskisson Syncline. These residuals indicate inadequate model geometry.

## 5.2. Rosebery–Lyell

The Rosebery–Lyell 3D model (Figure 11) extended the Rosebery Region model 30 km further south, encompassing the major Cu-Au mineral system at Mount Lyell and the nickel system at Avebury, for a total extent of 48 × 75 km (Figure 1A'). The geological units represented in the extended volume are largely the same as for the Rosebery region, and their geometry similarly defined explicitly from re-synthesis of the earlier models and structural interpretations [43,45]. Changes were made to the model's structural framework. For example, the boundary between the Tyennan Region and the Mount Read Volcanics/Dundas Trough ("Tyennan Margin") had been interpreted in the Statewide 3D model from potential field "worms" [45] as an east-dipping fault. In 1999, Pasminco Australia Ltd. (Exploration) produced a 3D model in the vicinity of the Rosebery mine indicating a west dip for the margin [45]. The latter interpretation was preferred for the Rosebery–Lyell model due to the ambiguity of "worm" sources and their susceptibility to interference in instances of multiple sources and complex geometry [55]. Additional changes were made to the geometry of the King River Fault (KRF) located in the southeastern sector of the model. An unpublished MRT report [56] proposed that the (KRF) is a major decollement fault with a listric west-dipping form based on the overturned nature of the western limb of the King River Synclinorium, which indicates overriding by a major thrust fault interpreted to be the KRF.

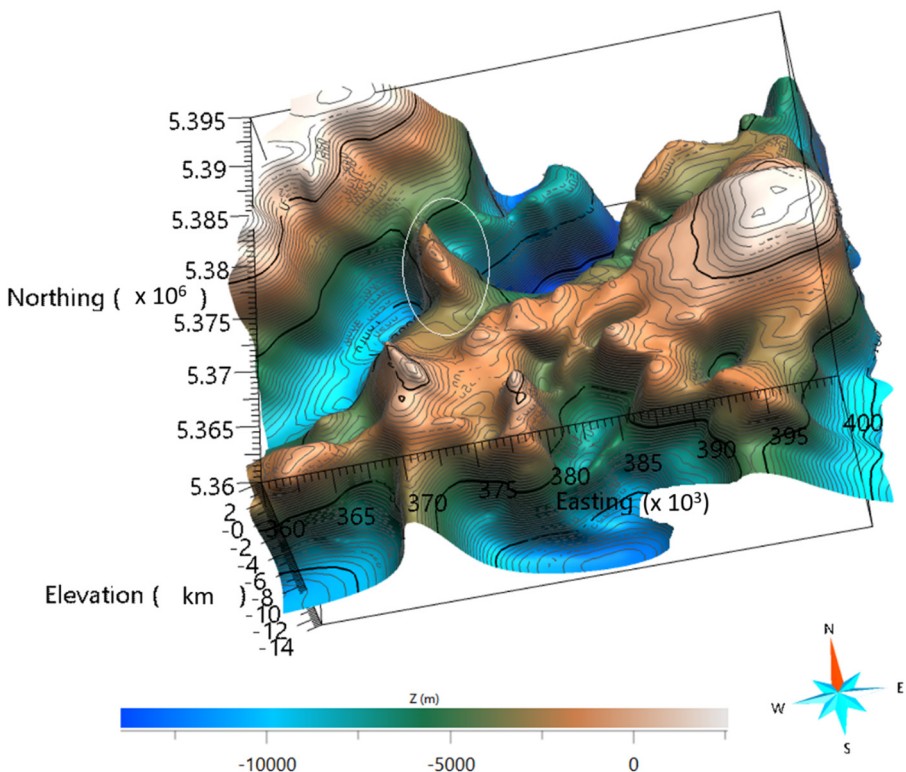

**Figure 10.** The geometry of the low-density Devonian granite was varied via inversion to account for a 10 mGal negative residual. This took the form of a spine underlying this region (white ellipse). Depth to this crest at its shallowest is estimated to be ~1000 m.

Substantial work was required to create 3D surfaces describing this complex terrane. The construction of geologically sensible stratigraphic and fault surfaces at the scale attempted was laborious and time-consuming. Following this, the model was discretized into 200 × 200 × 100 m voxels for geophysical modeling, resulting in ~10 million cells. An iterative series of magnetic and gravity inverse modeling tasks were undertaken using the methodology described in Section 5.1.1.

### 5.2.1. Magnetic Response

The magnetic response of the Rosebery–Lyell model, following homogeneous inversion, broadly accords with observed magnetic features, indicating plausible first-order geometry for the mafic–ultramafic complexes and Cambrian granites, which are the magnetically dominant units. However, both long- and short-wavelength features remaining in the post-inversion magnetic residual misfit (Figure 12) indicate unresolved geometric and intra-unit compositional variation, particularly with the ultramafic complexes, as was identified in the Rosebery Region.

These issues were investigated by further inversion steps allowing mafic–ultramafic unit volumes to vary both geometrically and heterogeneously (i.e., intra-unit susceptibility variation). The outcomes of this inversion run are illustrated in Figure 13, showing intra-unit compositional and geometric changes to the ultramafic geometry (i.e., addition of ultramafic material) west of the Huskisson syncline. This result is supported by inverted electromagnetic data (VTEM (Versatile Time Domain Electromagnetic), [54]) showing a formational conductor corresponding to the modified ultramafic geometry and the extension of the ultramafic complex under Cenozoic sediment cover [57]. Geologically, this can be explained with a wedged ultramafic thrust sheet in fault contact with the Eldon, Gordon, Owen, and Tyndall Group lithologies, (see Section 5.3.3 and Figure 17).

Previous modeling of the Rosebery region (Section 5.1.2) highlighted significant variations in apparent magnetization within mafic–ultramafic volumes. Results from heterogeneous inversion show at least three magnetically distinct units within the ultramafic lithologies (Figure 13). These were explicitly defined separately in subsequent modeling (see Section 5.3.1 and Table 2). Finer-scale features in the magnetic residual misfit are more confidently ascribed to volumes that are magnetically anomalous in a geological as well as geophysical sense.

**Table 4.** Eastern Terrane existing and model-informed mean susceptibility (SI $\times 10^{-3}$).

| Lithology | Existing [1] | Output [2] | 1 σ |
|---|---|---|---|
| Cenozic Basalt | >10 | 24.67 | 0.014 |
| Jurassic Dolerite | >20 | 26.17 | 0.007 |
| Parmeener Supergroup | 0.05 | 0.10 | 0.000 |
| Devonian Carboniferous Granites | 0.2–0.25 | 0.10 | 0.000 |
| Granodiorites | 0 | 0.30 | 0.000 |
| Magnetic Granodiorites | 6–8 | 7.51 | 0.002 |
| Mathinna Supergroup [3] | 0.15–30 | 0.20 | 0.000 |
| Scamander Formation [4] | | 0.10 | 0.006 |
| Lone Star Siltstone [5] | | 1.70 | 0.010 |
| Cambrian Sediments | 0.2–2.5 | 0.73 | 0.010 |
| Mafic–Ultramafic Complexes | 0–400 | 105.00 | 0.059 |
| Proterozoic Basement | 0–2 | 0.3 | 0.000 |

Note: [1] denotes existing values sourced from [10,33] and MRT's publicly available physical rock property database, [2] denotes GeoModeller[TM]—informed sensitivity statistics for mean susceptibility (SI $\times 10^{-3}$) from an ensemble of acceptable models, [3] denotes susceptibility value range for Mathinna Supergroup sourced from MRT's publicly available physical rock property database. [4,5] denotes model informed individual unit susceptibility for Mathinna Supergroup subunits.

### 5.2.2. Gravity Response

Bouguer gravity lows are associated with Devonian granite at or near the surface. These granite signatures can be clearly seen (Figure 8) in the northwest corner of the study area (the Meredith granite) and trending in a northwest to northeast direction in the northern half of the model. Longer-wavelength gravity highs represented by denser lithologies are located in both the southern and northern sections of the study area.

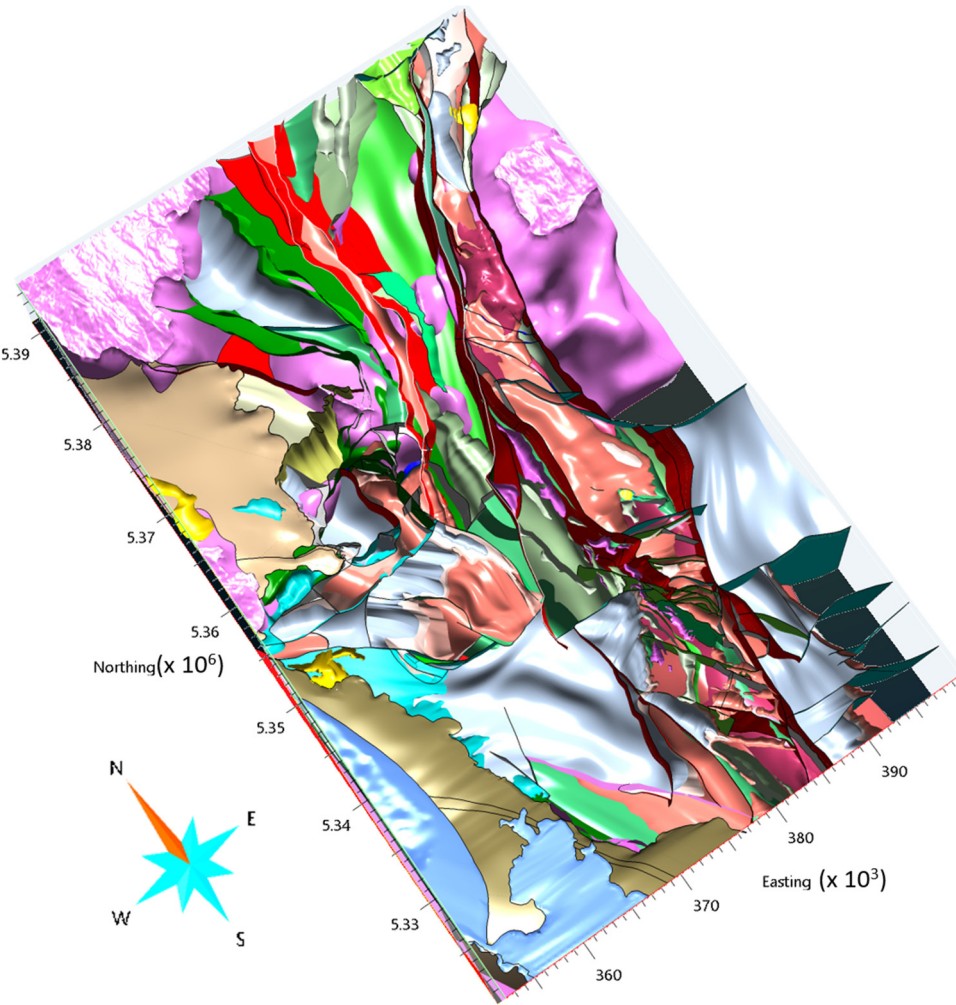

**Figure 11.** Three-dimensional schematic illustrating the complexity of the Rosebery–Lyell 3D geological model, Tasmania's most important mineral province. Lithological surfaces are represented by multicolored surfaces, granites (pink surfaces) and fault network architecture (red and blue surfaces). Macquarie Harbor (blue surface) shown in southwest corner of the model.

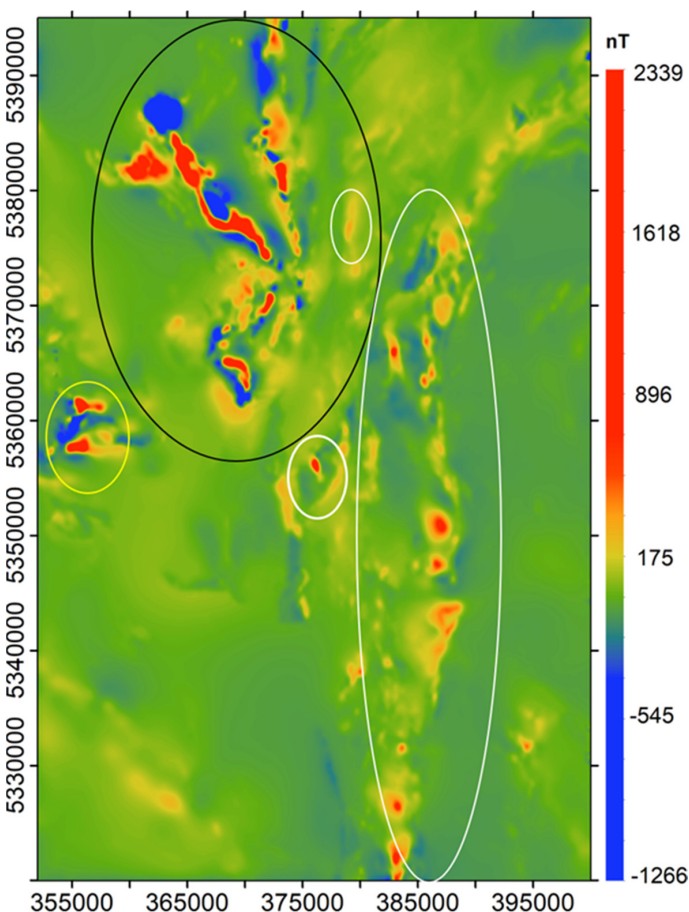

**Figure 12.** The TMI residual misfit of the Rosebery–Lyell reference model after homogeneous inversion (rms misfit 143.65 nT). Black and yellow ellipses represent broad-scale features associated with ultramafic complexes suggesting substantial compositional variation or inadequate model geometry. White ellipses highlight finer-scale features in the magnetic residual misfit and are more confidently ascribed to volumes that are magnetically anomalous in a geological as well as geophysical sense, such as the magnetite alteration system at Red Hills (central white ellipse).

Forward and homogeneous gravity modeling results showed a reasonable correspondence with the overall large-scale observed magnetic features including bulk unit physical rock property values. Figure 14 shows the gravity residual misfit after homogeneous inversion and geometry modification. Long-wavelength negative residuals (between 8 and 10 mGal) are located along the western boundary of the model. The authors of [42] suggested these to be associated with non-magnetic Cambrian granite with a density of ~2.60 kg/m$^3$. In contrast, long-wavelength positive residuals located in the southwest sector of the model are attributed to areas of poor gravity coverage and are most likely the result of inadequate model geometry. Longer-wavelength positive residuals located on the southern, western, and northern edge of the model are due to edge effects and may reflect regional de-trending and that padding algorithms are not entirely accounting for external source effects outside the model area.

### 5.2.3. Cooperative Potential Field Inversion

Geomodeller$^{TM}$ software was used for sensitivity modeling (see Section 3.2). The computational demands of the Bayesian approach, Markov chain, and Monte Carlo simulation used to derive model sensitivity were considerable given the resolution of the model voxel resolution 100 (x, y, z) to 200 m and consequently approximately 10 million model voxels. Model runs required an Intel Zeon-based supercomputer with upwards of 200 gigabytes of RAM and a run time of ~2 weeks in duration.

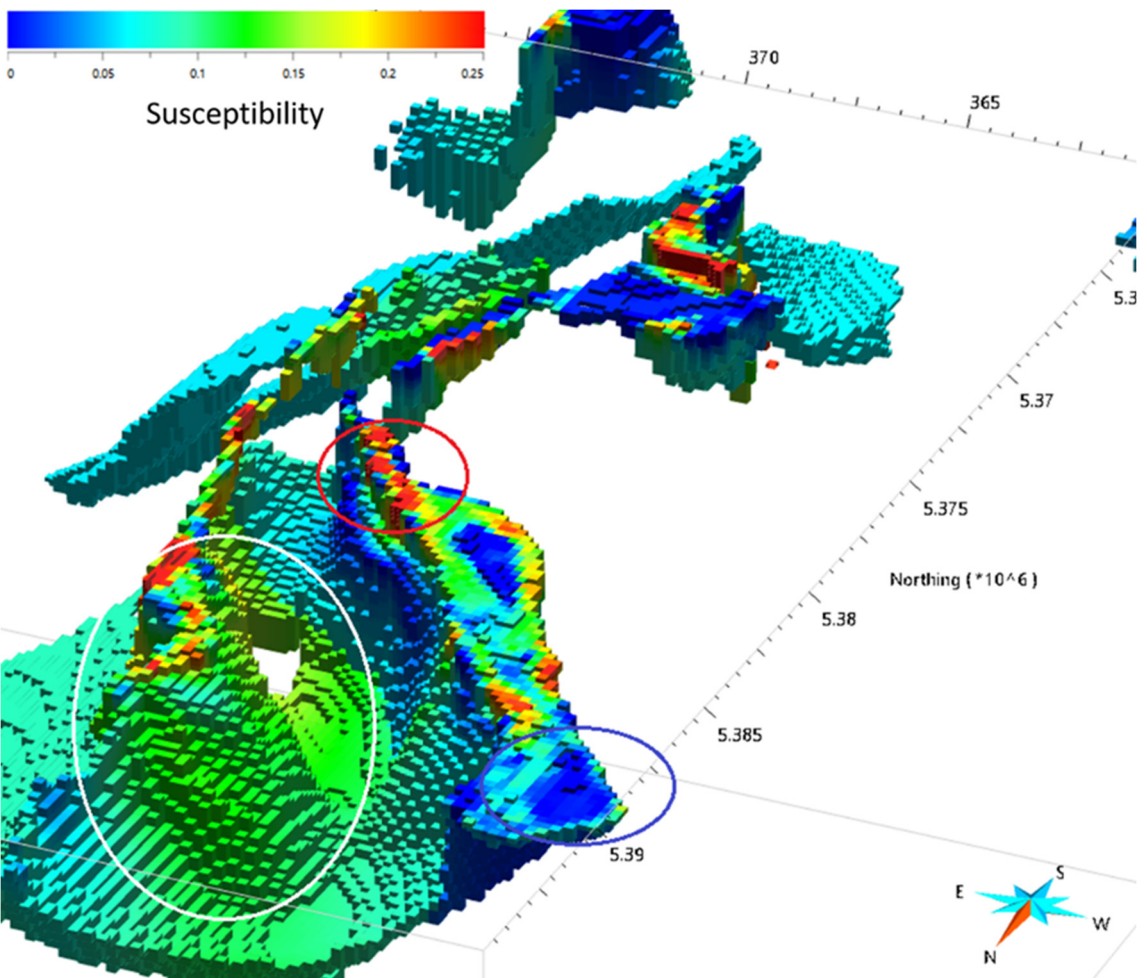

**Figure 13.** The voxel model representation of ultramafic units in the northern half of the Rosebery–Lyell 3D model after heterogeneous inversion. Note three magnetically distinct units. Black ellipse indicates intra-unit low-susceptibility region, white ellipse indicates medium susceptibility region, and red ellipse indicates a high-susceptibility region (SI).

The results exhibited problematic characteristics, particularly with the enforcement of constraints on the extent of departures of geological unit shapes from those of the initial model. As the geometric constraint parameters are defined in terms of the proportion of model cells, the complexity and very large size of the Rosebery–Lyell 3D model led to an inability to find a "sweet spot" allowing the proposal of an evaluation to operate appropriately across a wide range of unit volume sizes in the model. This resulted in either inordinate computational time calculating the geophysical response of geologically implausible models, or model inversion runs did not sufficiently explore the space of potentially acceptable solutions. Consequently, uncertainty characterization of the Rosebery–Lyell model was not concluded satisfactorily. These issues were, to a large extent, resolved for subsequent modeling by software improvements and applying the approach to smaller, more tractable models.

### 5.2.4. Outcomes

Notwithstanding the issues with sensitivity characterization, a granitic spine underlying the Huskisson syncline, with a contribution from adjacent serpentinized ultramafics, was a persistent feature in all model runs accounting for the Tributary Creek Gravity Anomaly. The characteristics of the TCGA were confirmed by the additional gravity data, mineral explorers had sufficient confidence to target a drillhole at potential mineralization expected in the vicinity of the modeled granite/serpentinite/carbonate contact [54]. The drillhole encountered serpentinized ultramafics throughout, consistent with the final

Rosebery–Lyell model, but unfortunately was terminated by technical problems at 582 m, still hundreds of meters above the predicted depth of the granite contact modeled to be at ~1000 m below the surface. The information from this drilling and its incorporation into follow-up modeling is described in Section 5.3.3. below.

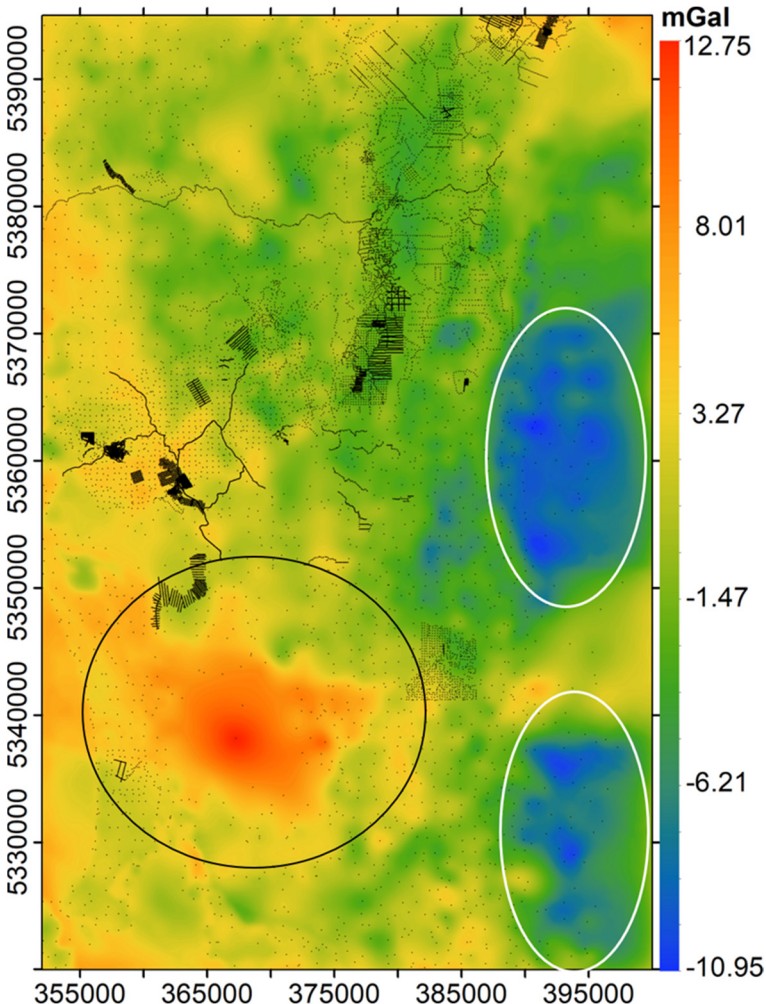

**Figure 14.** Forward model gravity residual misfit (rms misfit 2.53 mGal) of the Rosebery–Lyell reference model. Note longer-wavelength negative residuals (white ellipses) and positive residual (black ellipse) are indicated. Negative residuals indicate inadequate model geometry and require less dense material, possibly in the form of non-magnetic Cambrian Granite [42].

### 5.3. Rosebery–Pieman

Addressing the sensitivity characterization issues experienced with the size and complexity of the Rosebery–Lyell model, the Rosebery–Pieman model focused on a considerably smaller area, less than a third that of Rosebery–Lyell (Figure 1A″). The model was again constructed explicitly (Section 5.1.). While incorporating most elements of the Rosebery–Lyell 3D model, modifications were made to the Huskisson syncline to resolve structural consistency issues evident in the earlier phase of modeling. This included insertion of the Owen and Tyndall Group lithologies, which regionally lie unconformably over allochthonous mafic–ultramafic complexes. The reference model fault architecture and lithological surfaces are illustrated in Figure 15. In preparation for forward and inverse modeling, the geological unit geometry was discretized into 250 × 250 × 100 m voxels for geophysical modeling, resulting in ~1.7 million cells.

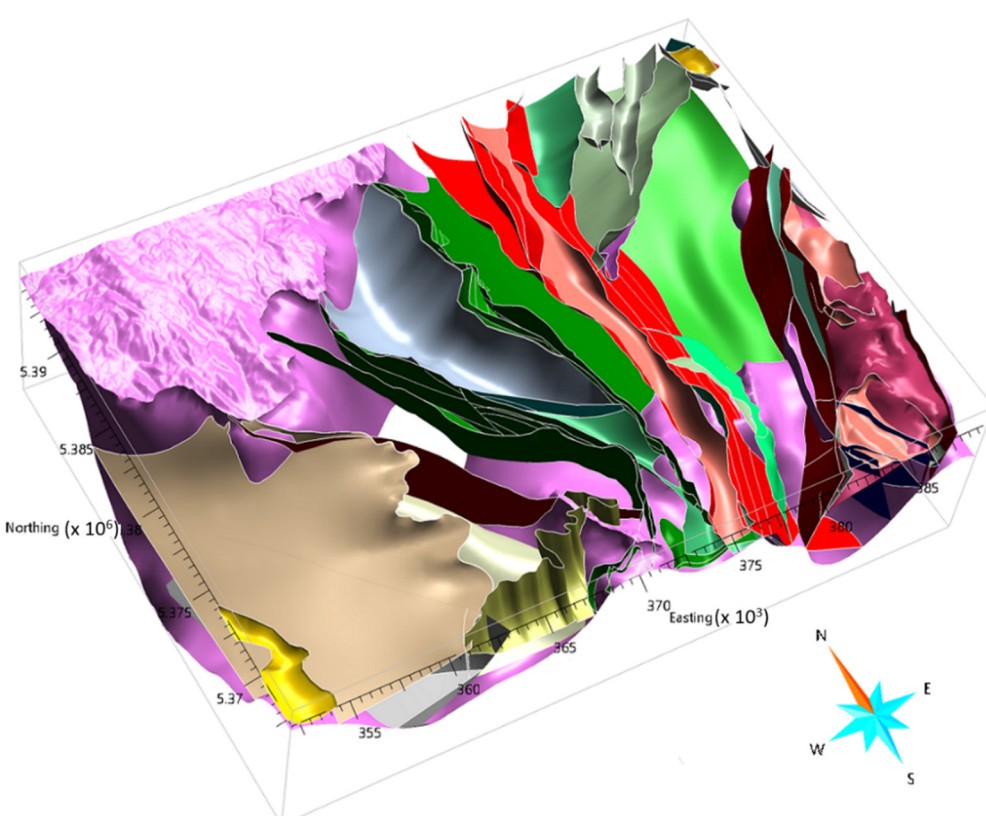

**Figure 15.** A 3D schematic illustrating the Rosebery–Pieman reference model. Lithological surfaces are represented by multicolored surfaces. Pink surfaces equal Devonian Granite and magenta surfaces equal Cambrian Granite. Red surfaces represent the fault network architecture.

### 5.3.1. Reference Model Magnetic and Gravity Modeling

Previous heterogeneous modeling of the Rosebery region (Section 5.2.1) highlighted significant variations in apparent magnetization within mafic–ultramafic volumes. Results show at least three magnetically distinct domains (Figure 13). We subdivided ultramafic volumes into three distinct units (see Table 2) and allowed each to vary geometrically. Further forward and homogeneous property gravity and magnetic modeling followed. Thus modified, the reference model response was sufficiently close to the observed magnetic and gravity data to proceed with refinement via cooperative inversion and stochastic sensitivity characterization.

### 5.3.2. Cooperative Potential Field Inversion

For this study, approximately 85 million acceptable models were made available for sensitivity analyses. Of these, approximately 32 million consisted of geological unit boundary changes and approximately 67.0 million consisted of physical rock property changes. Here, we focus on the probability threshold, a model sensitivity statistical measure, that records the lithology assigned to a voxel in at least 99% of the acceptable models satisfying the observed magnetic and gravity data. Additional statistical measures include the mean density and mean susceptibility, which are also derived from the accumulated accepted inversion proposals/models.

The gravity inversion converged first (approximately 500 million iterations), and the rms misfit stabilized at approximately 1.60 mGal (the burn-in point). This result demonstrates confidence in the spatial geometry of lithological units used in the reference model. Long-wavelength positive and negative residuals are present at the western, northern, and eastern boundaries of the model (Figure 16B) and may reflect regional de-trending and that padding algorithms did not entirely account for external source effects outside the model area. Long-wavelength positive residuals located on the western edge of

the model and underlying the Meredith granite and Huskisson syncline may be attributed to inadequate model geometry.

The magnetic inversion took longer to converge (approximately 1 billion iterations), with the misfit stabilizing at approximately 45 nT. This result is attributed to the tight geological property constraints used in the modeling, which restricted the number of proposals/acceptances. Figure 16A illustrates the final residual misfit. A long-wavelength positive residual in the southwestern corner of the model that is attributed to inadequate model geometry, suggesting a deep magnetic element in thrust contact with overlying Proterozoic rift sequences. Short-wavelength positive and negative residuals within mafic–ultramafic complexes, are probably geometric in origin or due to the inversion not accounting entirely for outlier magnetization values, possibly due to remanence. In contrast, short-wavelength residuals may be associated with mineral systems and are of direct exploration interest. For example, the red ellipse represents positive residuals proximal to the Renison Bell deposit while the black ellipse is associated with magnetic material within the Central Volcanic Sequence and is discussed in detail in Section 5.3.4.

### 5.3.3. Tributary Creek Gravity Anomaly

Figure 17 is an interpretive cross-section based on the most probable model sensitivity metric, which records the lithology most often assigned to a model voxel location from the ensemble of the ~85 million acceptable models. The sensitivity modeling confirmed the depth to the top of the granite surface underneath diamond drill hole (DDH) TCGA-01 at approximately 1300 m, approximately 300 m deeper than the reference model. Modeling also confirmed that ultramafic lithologies underlie the syncline at this location. These are interpreted to be folded thrust sheets in faulted contact with Gordon, Owen, and Tyndall Group lithologies.

Density measurements of DDH TCGA-01 core samples (ultramafic composition) range from 2.45 to 2.62 kg/m$^3$, largely in accordance with model estimates from 2.53 to 2.60 kg/m$^3$, after allowing for scaling to bulk volumes. Sample densities near the lower end of the measured range suggest total serpentinization of mafic–ultramafic protoliths. Susceptibility measurements from DDH TCGA-01 core samples range from 50 to 180 SI $\times$ 10$^{-3}$ are similarly consistent with model estimates (160–190 SI $\times$ 10$^{-3}$), especially after the likely natural remanent magnetism contribution is considered.

Carbonates become slightly more common approaching the bottom of DDH TCGA-01, with magnesite being the dominant species identified both by HyLogger$^{TM}$ (Figure 18) and XRD analyses. They occur in broken core and on joint/shear faces. A possible mechanism for the formation of magnesite could be the convection of meteoritic waters induced by a nearby magmatic source. Another possibility is low-temperature hydrothermal fluids from a magmatic source encountering limestone, which would enhance the $CO_2$ concentration and magnesite dissolution. The nearby Gordon Group limestone is a candidate $CO_2$ source and the shear zones observed within the drillhole could act as fluid conduits, with magnesite precipitating as a result of a decreasing fluid temperature. The nearby Devonian granitoid is a plausible magmatic fluid source.

### 5.3.4. Rosebery North

A positive residual magnetic anomaly emerging from the earlier modeling phases occurs north and northeast of the Rosebery deposit (see Figure 16). Figure 19 is an east–west section intersecting the residual magnetic anomaly and represents the probability threshold sensitivity metric. Lithologies include the prospective Western Volcano Sedimentary Sequence and CVC. The Rosebery polymetallic ore deposit occurs along the strike at this stratigraphic position, mainly to the south. In response to the residual magnetic anomaly, the inversion placed, with high probability, a unit at a depth of 2 km with petrophysical properties similar to Cambrian Granite. Cambrian granite has not previously been thought to exist west of the Henty Fault Zone. The nearest extant drill holes to the proposed intrusion (RED 87, 345R-D1, and 345R-D2) intersected magnetite–pyrite vein networks and

intrusive dykes with granodioritic composition similar to Cambrian granite elsewhere in the region [59], consistent with the position of the body being suggested by the modeling. A volume of non-magnetic anomalously low-density material emergent within the Central Volcanic Complex between two buried Cambrian granitoid bodies has properties consistent with K-feldspar alteration of felsic volcanics. Materials with similar physical rock properties have been noted in association with Mt Lyell-type copper mineralization within 1–2 km of Cambrian granite [59], suggesting potential for such a mineral system to exist here at explorable depths.

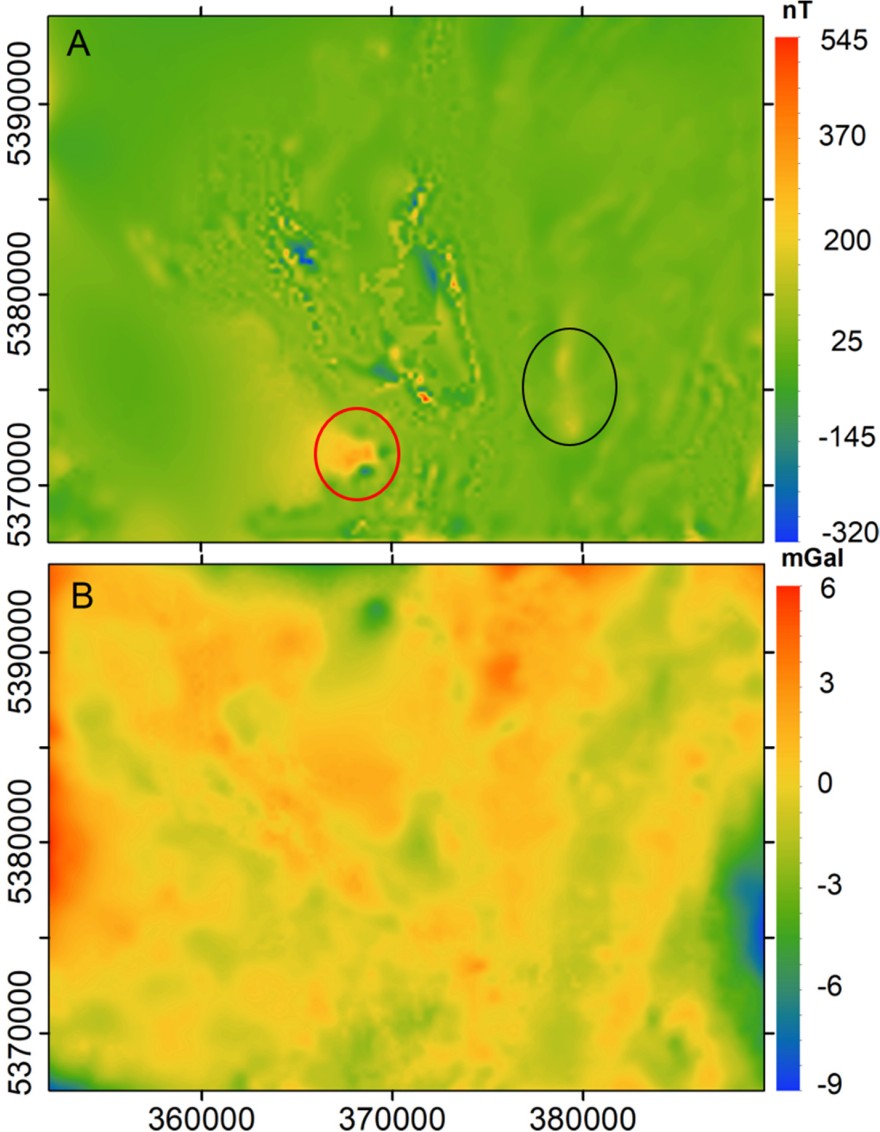

**Figure 16.** (**A**) The TMI residual misfit of the Rosebery–Pieman 3D model after cooperative inversion (rms misfit 41.47 nT). Ultramafic complexes were modeled as 3 distinct units due to their substantial compositional variation. Red ellipse represents positive residuals proximal to the Renison Bell deposit while the black ellipse is associated with magnetic material within the Central Volcanic Sequence. (**B**) The gravity residual misfit. The rms misfit is 1.56 mGal.

*5.4. Northwest Tasmania*

Three-dimensional models of NW Tasmania were developed as part of a PhD project co-supervised by the authors [15,60], using similar methods to those described above. The more extensive of the two models featured here covered the entirety of Northwestern Tasmania (Figure 1B).

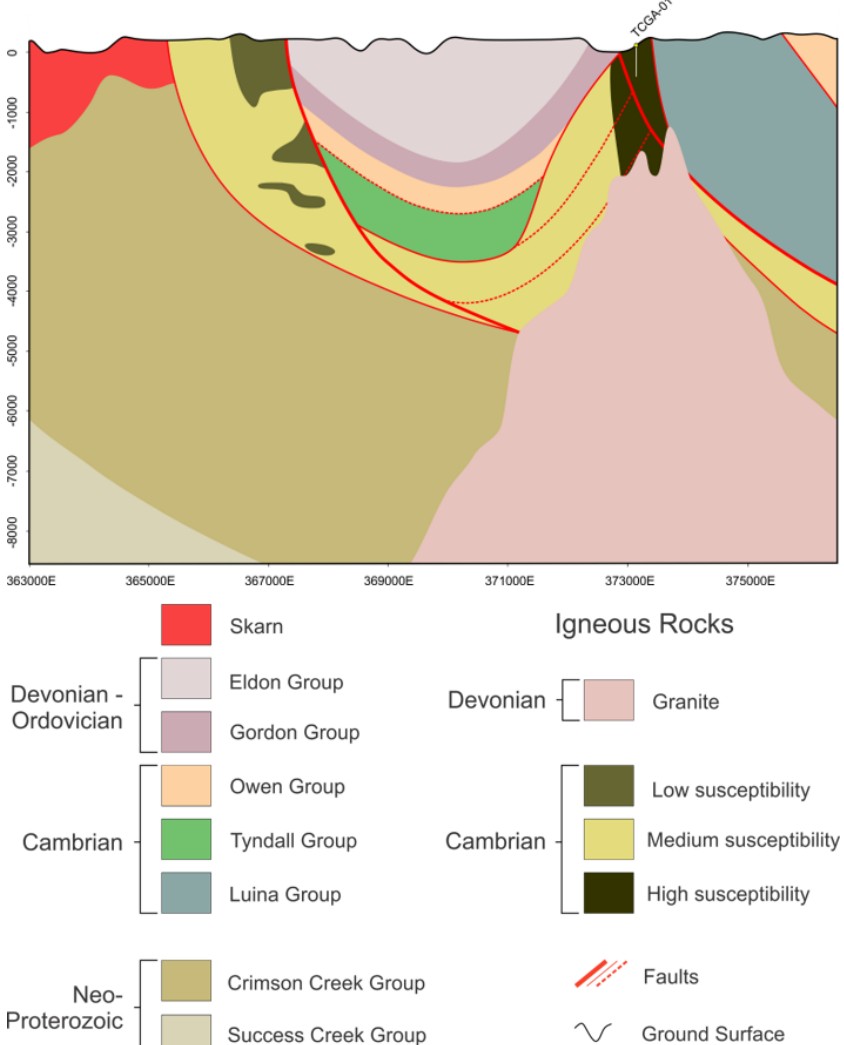

**Figure 17.** Interpretative east–west cross-section (Northing 5380723) through the Huskisson Syncline based on the most probable model sensitivity metric from cooperative inversion.

The potential field modeling [15] resulted in a number of improvements to the 3D geological understanding of the region. Combined gravity and magnetic signatures indicate the Housetop Granite as relatively thin (≤5 km thickness), with a mafic–ultramafic complex near its base at a depth of ~3–6 km, similar to the model of [61]. This granite form contrasts with other plutons in the region, such as the Heemskirk and Meredith Granites, whose gravity signature shows them to be more deeply rooted, in excess of 10 km, as well as being connected at depth, extending the area of granite-related mineralization potential. The Housetop Granite may have been truncated by a subsurface thrust fault, though this would present challenges to the current structural understanding of the Devonian granites in NW Tasmania being largely post-tectonic.

An entirely concealed granite intrusion was also inferred beneath the eastern Rocky Cape Group on the basis of an 8 mGal residual gravity low [15] (p. 23). No direct geological indications of this intrusion have yet been identified, nor are they necessarily expected (its depth being no less than 3 km); thus, its age is undetermined but possibly Neoproterozoic or Devonian. Another negative gravity residual in the southern sector of the model was tentatively interpreted as a southward extension of subsurface Cambrian granite [15]; however, its non-magnetic character marks it as dissimilar from the dominant Cambrian granitoid phases. An isolated intrusion of Devonian granite is another possibility.

### 5.5. Heazlewood–Luina–Waratah

The Heazlewood–Luina–Waratah region (Figure 1C) hosts several significant mineral systems. The 20 × 20 km model developed by [62] encompasses Proterozoic metasedimentary rocks, Cambrian allochthonous mafic–ultramafic complexes (including the Luina Group), Ordovician–Devonian sedimentary rocks, and Devonian granitic intrusives [62] (p. K16). The area also comprises the eastern zone of the Arthur Metamorphic Complex, which consists of various Proterozoic protoliths metamorphosed during the Cambrian Tyennan Orogeny [63].

As with the broader Northwest Tasmania modeling, the potential field inversion workflow resulted in significant modification to earlier conceptions of subsurface Devonian granite geometry. In the south, the Meredith batholith is at least 10 km thick, while the northern edge of the batholith plunges steeply. Granite cupolas at shallow depths (less than 2 km) were postulated to account for apparent residual gravity anomalies in the northeast.

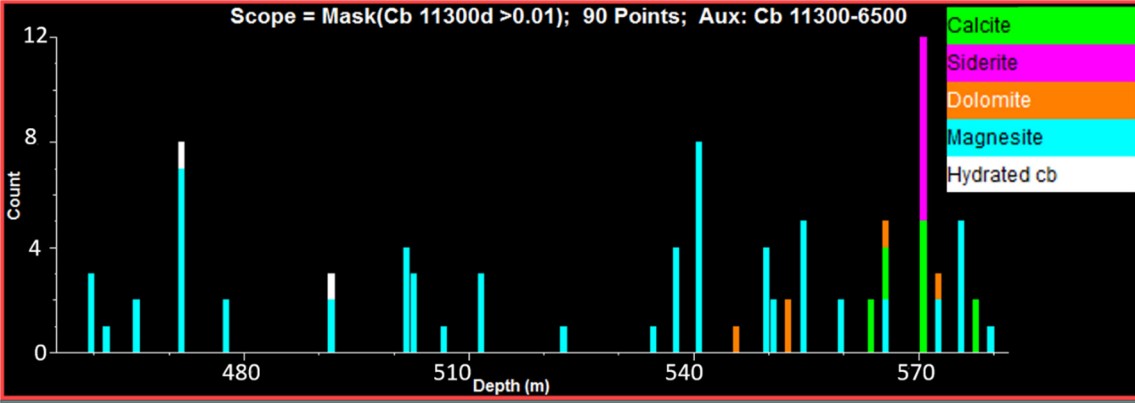

**Figure 18.** Carbonate abundances from HyLogger[TM] scanning of DDH TCGA-01, showing magnesite (aqua) as generally dominant, with other species including dolomite (orange), calcite (green), and, notably, siderite (magenta) becoming more common approaching the bottom of the drill hole.

The novel interpretation of an ultramafic complex linking the Heazlewood and Mount Stewart ultramafic complexes in the southwest beneath sedimentary rocks associated with the Bell Syncline implies a greater volume of ultramafic material in the Cambrian successions and points to a larger obducted component than previously thought [63] (p. K23). This interpretation was subsequently supported by the discovery of ultramafics outcropping in the area [64]. The Heazlewood ultramafic complex was shown via a sensitivity study to have a thickness of up to ~4 km, whereas the Mt Stewart ultramafic complex and the contact aureole surrounding the Meredith granite to the north are thinner (approximately <2 km; [60]). The contact zone of the Meredith Batholith and the Mt Stewart mafic–ultramafic complex extends to a maximum depth extent of 2 km. Both the newly inferred granite cupolas and ultramafic complexes are potential targets for Avebury-style Ni and other mineral systems. Heterogeneous property inversions showed that the Heazlewood River mafic–ultramafic complex is characterized by highly variable density and magnetic susceptibility [60]. Areas of poor to nonexistent gravity coverage meant that several areas in the model are of low confidence or unresolved. These locations include the western edge of the Heazlewood River ultramafic complex and north of the Mt Stewart ultramafic complex.

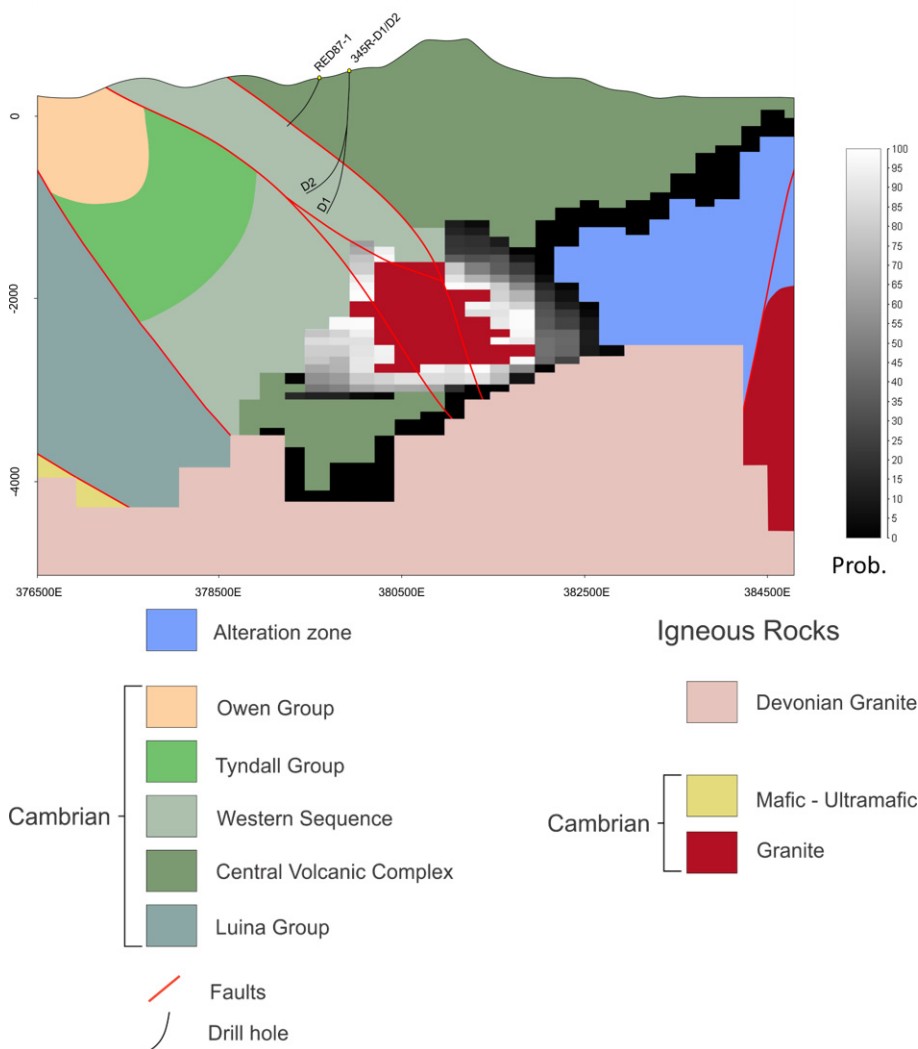

**Figure 19.** East–west cross-section intersecting residual magnetic anomaly at N 5377182 and illustrating the most probable threshold model geology. Greyscale legend shows the probability of Cambrian Granite and model-informed alteration zone (blue).

## 6. Eastern Tasmanian Terrane (ETT) 3D Models

Three 3D geological models of mineralized areas in the ETT have been constructed covering the Lebrina, Alberton–Mathinna, and Scamander areas (Figure 2). The simpler geology of the ETT compared to the WTT, lent itself to trialing different modeling approaches.

### 6.1. Mineral Systems

Tin and orogenic gold are the predominant mineral systems in the ETT. Sn is known to be closely associated with Devonian granite intrusion, whereas the origin of Au (in particular, the role of granitoid intrusion, if any) is far less certain, with structural controls at a range of scales. Historically, economic Au has been hosted mainly by quartz veins.

### 6.2. Lebrina

The Lebrina model was built entirely by implicit means, whereby surfaces bounding unit volumes were interpolated from geological observations (stratigraphic and fault contacts, dips, and strikes) via prior geological knowledge encoded in a matrix of rules defining the relative timing of all model components [10]. As a useful model can thus be constructed with far fewer user-entered points than explicit modeling, this makes it

much easier to modify the model in response to indications from subsequent geophysical modeling or emergent 3D geometric plausibility.

The upper several kilometers of Lebrina volume with extent 49.5 km (west to east) by 59.25 km (north to south) and depth of ~11 km (Figure 2A) mainly consist of metasedimentary Mathinna Supergroup country rock intruded by late Devonian–early Carboniferous granites and granodiorites, which outcrop in the eastern portion of the area. An allochthonous Cambrian mafic–ultramafic complex is thought to underlie much of the area, largely on magnetic grounds [33], while the southwestern half is covered by Jurassic dolerite and up to a few hundred meters of Permo-Triassic sediments (Parmeener Supergroup). Significant gold has been won historically from several goldfields across the Mathinna Supergroup and associated alluvial deposits.

### 6.2.1. Cooperative Geophysical Inversion

Litho-constrained alternating stochastic gravity and magnetic inversion described in Section 3.2 was undertaken. Constraints consisted of surface-mapped outcrops, structural information (e.g., dips), and shallow drilling information. The most notable result was a considerable expansion of granitoid beneath the outcropping Mathinna Supergroup in the east and, to a lesser extent, the center of the model region (Figure 20). This finding is, however, subject to considerable uncertainty due to low-density contrast between the predominantly granodioritic composition of the proposed extended batholith and the Mathinna Supergroup metasediments. Moreover, some lower-density, more granitic elements of the modeled intrusion are likely to be spurious where they coincide with topographically high areas. Here, residual gravity lows (Figure 21) giving rise to the putative subsurface granite cupolas were later found to be the result of inadequate terrain correction applied to the observed gravity data [39]. Relatively high uncertainty is also associated with the southwestern quadrant of the Lebrina model, where extensive magnetic, dense dolerite cover masks the gravity and magnetic responses of underlying units; nevertheless, a Cambrian sedimentary unit is predicted to occur beneath the Parmeener Supergroup west of the Tamar River.

### 6.2.2. Second Phase Implicit Modeling

A comprehensive revision and refinement of the Mathinna Supergroup's stratigraphy and structure [32] was incorporated into the Lebrina model via a second phase of implicit modeling as implemented in GoCAD/SKUA$^{TM}$ software. These Mathinna Supergroup subdivisions were not included in the preceding geophysical inversion as they have little density and magnetization contrast. An internal fault network was built first using structural information gleaned from outcrop data, expressed as an array of cross-sections. Stratigraphic surfaces were then added. The resulting internal volumes were then used to replace the homogeneous Mathinna Supergroup volume in the geophysical model (Figure 22).

No influence of the fault network is apparent on the form of the modeled granitoid intrusions, consistent with the latter being largely post-tectonic. Granodiorite is seen to extend continuously in the subsurface connecting isolated outcrops in the Lisle and Panama goldfields, coming within a kilometer of other gold occurrences in the Denison and Lebrina goldfields. Further extension of a subsurface granite and granodiorite spine to the NNW under Cenozoic sediments might indicate additional gold potential in this area; however, the genetic association between granitoid and gold in this region, if any, remains unclear [33].

### 6.3. Alberton–Mathinna

The Alberton–Mathinna area (Figure 2B) covers the ETT's most economically significant style of gold mineralization, occurring mostly within quartz veins along a trend extending NNW–SSE for tens of kilometers [36,65]. The regional geology is similar to that at Lebrina, with the Mathinna Supergroup lithologies intruded by Devonian granitoids.

Geological and geophysical modeling of this region is described by [10]. This study indicated a closer spatial association between some of the larger known gold accumulations and subsurface granite than previous interpretations, and hence the possibility for a greater role of magmatism in mobilizing mineralizing fluids. Features emergent in details of the modelled granite surface include a cupola in the vicinity of the Mathinna goldfield (the largest in NE Tasmania), and granite intruding the fault network underpinning the Ringarooma United deposit at a depth of approximately 400 m [10] (p.T537).

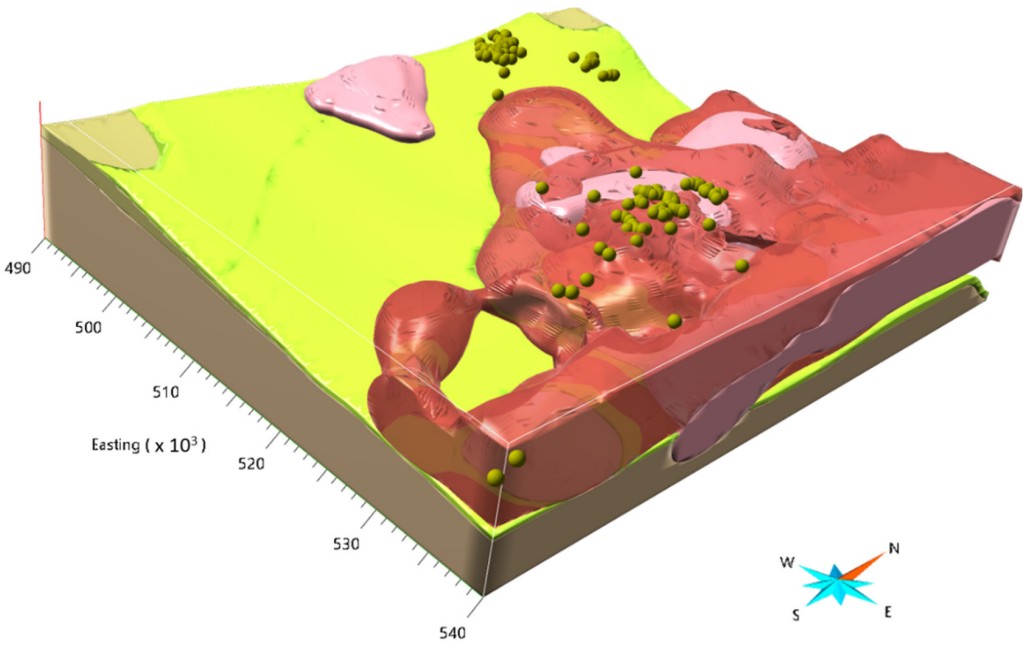

**Figure 20.** The Lebrina 3D model showing granite (pink) and granodiorite (salmon) volumes intruding ultramafic (light green) and basement (tan) lithologies. Note gold occurrences (gold spheres) proximal to granitic cupolas.

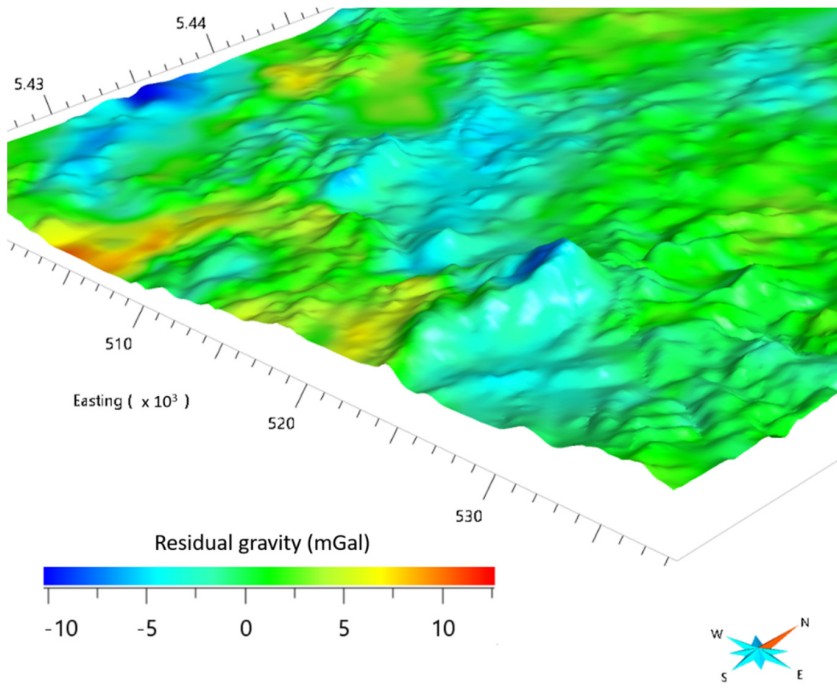

**Figure 21.** Gravity map draped over DEM. Note that gravity lows coincide with topographic highs resulting from inadequate terrain correction [39].

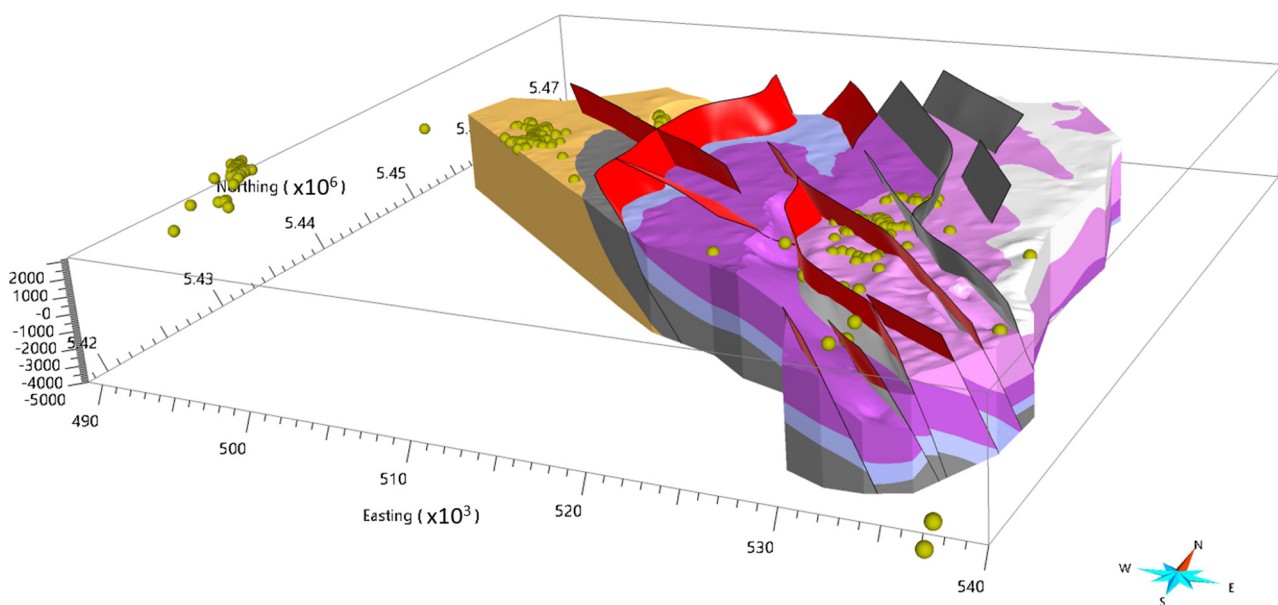

**Figure 22.** Second phase of implicit modeling for the Lebrina 3D model using GoCAD/SKUA$^{TM}$. Further refinement of the Mathinna Supergroup stratigraphy and associated fault (red and grey surfaces) network relationships (e.g., thrusting). From youngest, Sideling Sandstone (white), Lone Star Siltstone (pink), Retreat Formation (purple), Yarrow Creek Siltstone (light blue), Turquoise Bluff Slate (dark grey), and Industry Road member (tan).

### 6.4. Scamander

The Scamander area (Figure 2C) is an example of a zoned mineral field, with mineralogy interpreted as a function of distance from the interpreted source granite intrusion [66]. W-Mo occurs adjacent to outcropping granite in the NE extending south-eastwards through Sn (Great Pyramid) and Cu (Orieco) to Ag-Pb-Zn near the coast. Zonation is hypothesized to reflect a gently dipping subsurface extension of the granite beneath the mineral field [67], which is broadly confirmed by gravity modeling [68]. The country rock intruded by the granite is the upper Mathinna Supergroup, comprising Siluro-Devonian, Lone Star Siltstone and Devonian Scamander Formation.

#### 6.4.1. Model Construction

A reference model was constructed using the explicit modeling method discussed in Section 3.1., via serial cross-sections of the area following [69]. The model was discretized into 250 × 250 × 100 m cells for forward and inverse modeling using the GOCAD® potential field module (VPmg code; [5]). Unit property estimates were derived from MRT's petrophysical database.

#### 6.4.2. Geophysical Modeling

An approach iterating magnetic/gravity forward, homogeneous (single property value for an entire unit), and geometry inversion was taken. For geometry inversion, the Devonian Granite was allowed to vary to account for long- and short-wavelength density residuals. Changes to the starting granite surface included prominent pronounced cupolas proximal to major faults and deposits (Figure 23). Further homogeneous inversion was undertaken with this new granite geometry to further constrain single property values for each lithology. Results show a reasonable correspondence with the overall large-scale observed gravity features including bulk unit physical rock property values (Table 3).

Magnetic modeling was difficult due to all model units being largely non-magnetic. Those magnetic anomalies that are observed are associated with thin quartz dolerite dykes intruding along NE-trending faults, and anomalous relatively small features. The latter are ascribed to (monoclinic) pyrrhotite accumulations, which are known to be associated with several of the mineral occurrences [67]. Indicative geometry of these sources was defined

by allowing the susceptibilities of the Lone Star Siltstone and Scamander Formation to vary upwards heterogeneously (Table 4), with results outlined below.

### 6.4.3. Cooperative Geophysical Inversion

Approximately 117 million acceptable models were generated for sensitivity analysis. Of these, approximately 6 million consisted of geological unit boundary changes and approximately 111 million consisted of physical rock property changes. Statistical measures used for this study include the probability threshold, which records the lithology assigned to a voxel in at least 99% of the acceptable models which satisfy the observed magnetic and gravity observations. Additional statistical measures include the mean density and mean susceptibility, which are also derived from the accumulated accepted inversion proposals/models.

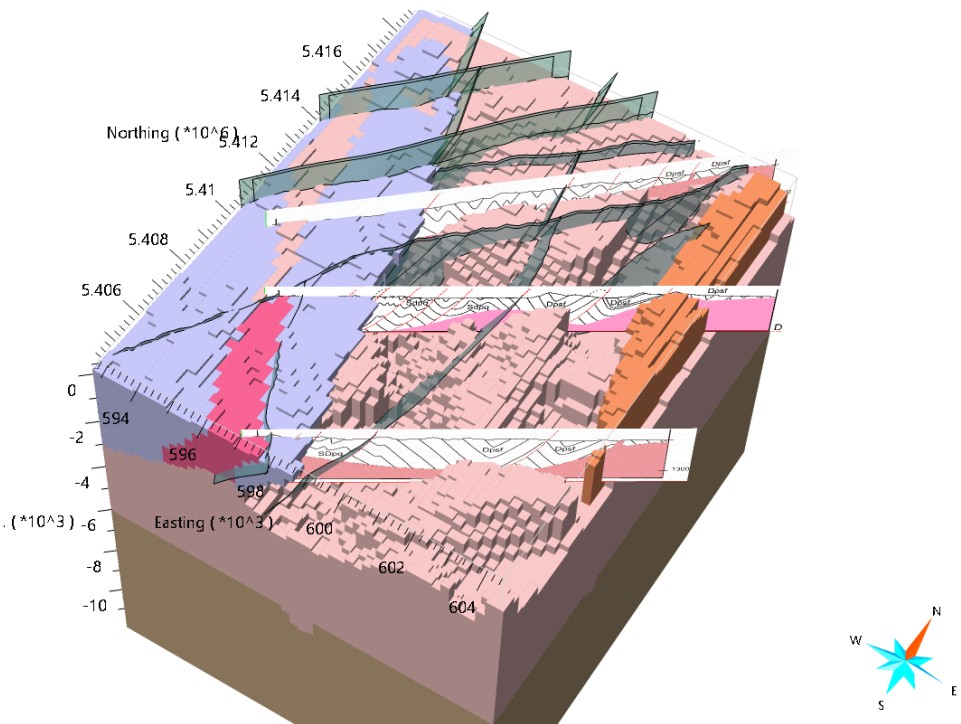

**Figure 23.** Discretized voxet reference model of the Scamander 3D model illustrating interpretative cross-sections, granite surface (pink), and fault network (transparent turquoise grey).

The gravity inversion converged first (approximately 25 million iterations), with the rms misfit stabilizing at approximately 0.3 mGal, close to the noise estimate of the observed data given the model resolution. Multiple small-scale short-wavelength positive and negative features in the residual gravity misfit (Figure 24A) indicate departures from bulk mean unit properties that may arise from alteration or other processes associated with mineral systems, and thus present targets for exploration follow-up. Edge effects at the eastern boundary of the model may reflect regional de-trending and that padding algorithms are not entirely accounting for sources located just outside the model area. A positive residual was associated with the Catos Creek dyke, suggesting that this granodiorite may be more dense and thus possibly more mafic in composition than had been ascribed.

The magnetic inversion took longer to converge than gravity (approximately 50 million iterations), with the misfit stabilizing at approximately 3 nT. Figure 24B illustrates the residual misfit, which was negligible, except for a few voxels with outlier positive residuals, which the model could not account for. These may represent additional volumes of anomalously magnetic (possibly pyrrhotitic) material in the Mathinna Supergroup above and beyond the more magnetic sub-population allowed by the a priori defined bimodal magnetic susceptibility distribution for these units. The most prominent instance

of this on Figure 24B (~2 km east of center) is the North Scamander Pb-Cu-Ag-Zn prospect, where massive magnetite (up to 40% by volume) has been intersected by drilling at a DDH depth of ~200 m. More moderately magnetic inversion voxels shown at 599900E and 602000E on the 5413500N east–west cross-section (Figure 25B) correspond, respectively, to disseminated and vein magnetite drilled at depth in the Great Pyramid Sn deposit (SPG 1A, with highly variable susceptibilities, [67]), and a magnetic anomaly extending over 1.5 km NNW, where disseminated pyrrhotite has been intersected by drilling (CR DH1).

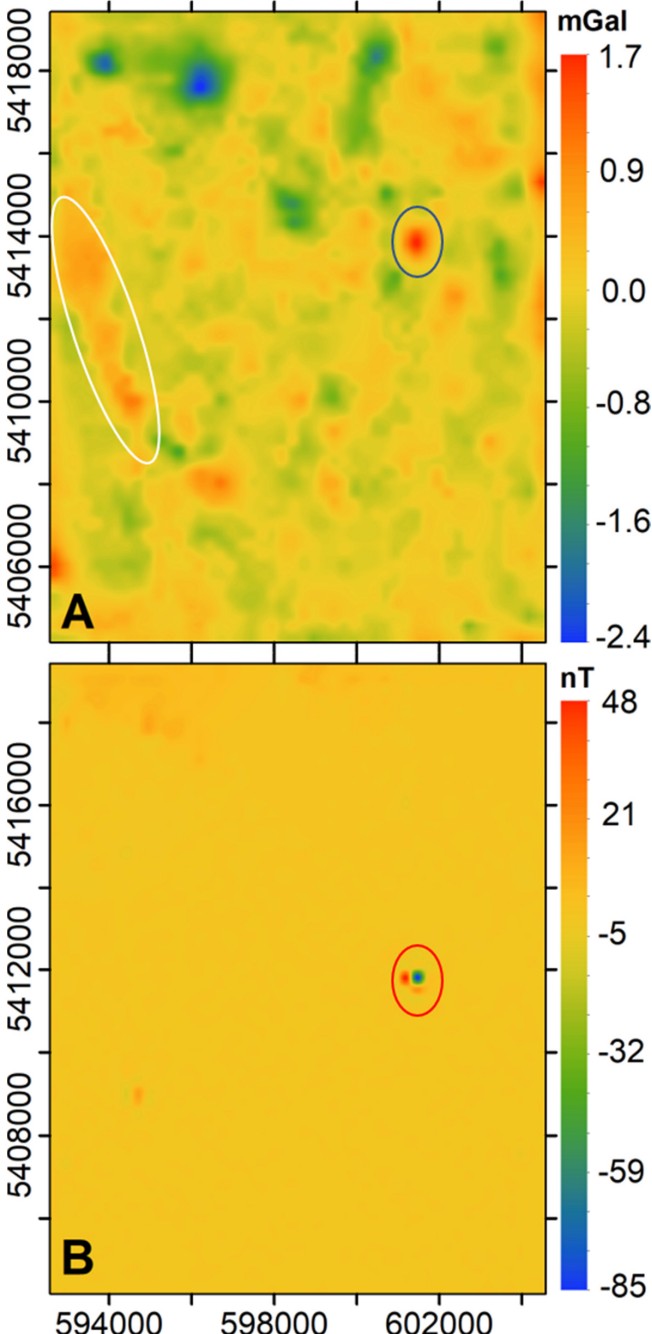

**Figure 24.** (**A**) The gravity residual misfit after cooperative inversion for the Scamander 3D model. The rms misfit stabilized at around 0.3 mGal, which is very low. Black and white ellipses highlight positive residuals associated with the Orieco copper prospect and Catos Creek dyke. (**B**) The TMI residual misfit with the rms misfit stabilizing at around 3 nT. The red ellipse represents outlier residuals associated with the North Scamander Cu, Pb-Zn-Ag prospect. The deposit is strongly magnetic, consisting of vein-hosted sulfide minerals with massive magnetite and pyrrhotite [67].

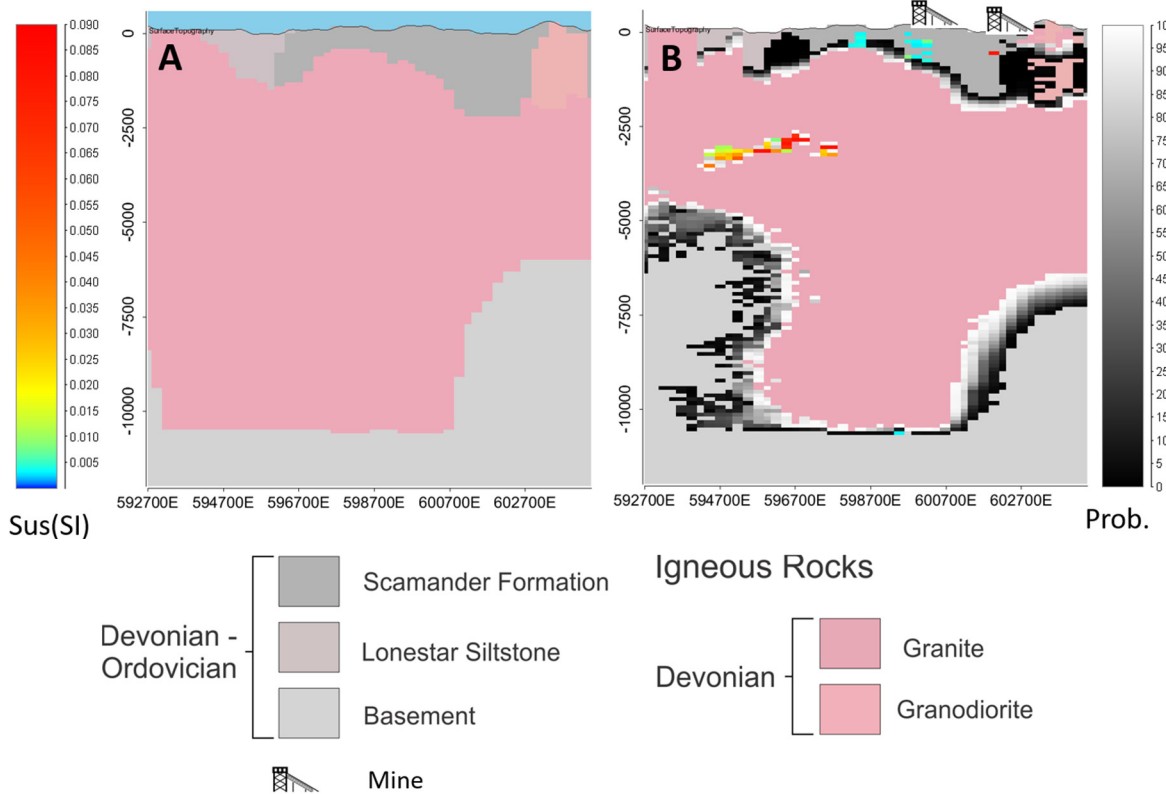

**Figure 25.** (**A**) East-west cross-section illustrating the reference starting model geology and intersecting the Great Pyramid Sn deposit at N 5413500. (**B**) illustrates the most probable threshold model geology and includes the probability of Devonian Granite and mean susceptibility informed by sensitivity modeling.

### 6.4.4. Inversion Analysis—Great Pyramid Tin and Other Deposits

Figure 25A is an east–west section of the reference model passing through the Great Pyramid Sn deposit. Compared to the reference model, the inversion (Figure 25B) placed, with high probability, a less voluminous granite. Depth to the top of the prominent cupola remains at approximately 400 m (beneath that of the deepest drilling to date at Great Pyramid, 348 m), suggestive of being the source of mineralizing hydrothermal fluids responsible for the Great Pyramid deposit. The North Scamander and Orieco copper deposits are also proximal to nearby granite cupolas, supporting the granite's role in their genesis. Overall, the inversion result supports the hypothesis of granite-derived fluids as the most likely source of metals, with the temperature of the fluid and distance from granite controlling the mineralogy of the deposits in the Scamander Mineral Field.

The emergence of granite roots extending some kilometers downwards is a more questionable outcome of the inversion, possibly indicating overfitting of the gravity data or insufficient density contrast with the underlying basement material. A high degree of uncertainty is placed at the boundary between Devonian Granite and Mathinna lithologies. High levels of uncertainty are also associated with the geometry of the Scamander Tier Dyke, which the inversion renders less voluminous than in the reference model. This is attributable to the low density contrast between Mathinna Supergroup meta-turbidites and the granodiorite dyke.

### 7. Conclusions

The release of the Tasmanian Statewide 3D model in 2003 coincided with a substantial increase in mineral exploration activity in Tasmania. A number of a targets were explicitly identified, the most notable being a VHMS target located under Cenozoic basalt cover north of the Hellyer deposit. The Statewide 3D model also formed the basis for a comprehensive geospatial assessment of the State's mineral potential across 45 mineral system models [70].

In 2011, MRT embarked on the development of next-generation high-resolution regional 3D models of Tasmania's Western and Eastern Terranes. These models incorporated higher levels of stratigraphic and structural detail and were constrained by full three-dimensional magnetic and gravity modeling. The first of these to be constructed was the Rosebery Region 3D model, covering Tasmania's most important mineral province. This modelling identified a significant negative gravity residual (TCGA) near the prospective southwestern Huskisson Syncline. This stimulated further exploration and research interest in the form of a gravity survey and drilling program, including a series research projects (The Northwest Tasmania and Heazlewood–Luina–Waratah 3D models).

Derivative models of the Rosebery Region (i.e., the Rosebery–Lyell and Rosebery–Pieman 3D models) were constructed incorporating statistically generated sensitivity characterization to map confidence in the geometry of geological units at depth. Results confirmed the depth to the top of the granite surface underneath the TCGA at approximately 1300 m. An additional finding, placed with high probability, is a magnetic body with petrophysical properties similar to Cambrian Granite at a depth of approximately 2 km underlying Mt Black. This implies the potential for a Mt Lyell-type mineral system.

The Eastern Terrane 3D models (Lebrina, Mathinna–Alberton, and Scamander) successfully delivered new insights particularly into the subsurface geometry of the granitoids that extensively intrude the region. While the models in themselves cannot be conclusive, spatial relationships were observed between the intrusions and known orogenic gold and base metal mineralization. The Lebrina model was first built entirely by implicit means. The most notable result was a considerable volume of granitoid beneath out-cropping Mathinna Supergroup in the east. However, residual gravity lows giving rise to the putative subsurface granite cupolas were later found to be the result of inadequate terrain correction applied to the observed gravity data. A comprehensive revision and refinement of the Mathinna Supergroup's stratigraphy and structure was incorporated into the Lebrina model via a second phase of implicit modeling to visualize in 3D the Mathinna Supergroup's stratigraphy as mapped by [32].

In the case of the Scamander model, results from sensitivity analysis produced a significantly more detailed granite model when compared with previous models. Among the new features to emerge are cupolas in the vicinity of the Great Pyramid, Orieco, Lutwyche, and the North Scamander prospects. This result supports a granitic origin of metals in the Scamander mineral field, and points to the likely role of permissive structures in the Mathinna Supergroup country rock for localizing mineralizing fluids as well as controlling granite intrusion.

Our experience shows that deriving insights from 3D modeling can be hindered by models that are too large or too detailed to be interrogated easily, especially when modeling techniques do not readily permit significant geometric changes. The most effective 3D modeling workflow for insights into mineral exploration is that which facilitates the rapid hypothesis testing of a wide range of scenarios whilst satisfying the constraints of observed data. Future modeling platforms—for example the LOOP project [71], if successful will enable rapid hypothesis testing in the form of interoperable, integrated, and probabilistic 3D geological and geophysical modeling.

Mineral systems operate through time and in three dimensions. Discovery rates will be improved through better modeling of the geometry of geological controls on mineralizing fluids. MRT continues to develop high-resolution regional 3D models that can be used by exploration companies to make more informed exploration decisions.

**Author Contributions:** Conceptualization, D.B., M.D. and A.R.; Formal analysis, D.B.; Investigation, D.B.; Methodology, D.B. and M.D.; Project administration, D.B. and M.D.; Writing—original draft, D.B.; Writing—review & editing, D.B., M.D., A.M., M.C. and A.R. All authors have read and agreed to the published version of the manuscript.

**Funding:** This research received no external funding.

**Data Availability Statement:** The data used in this study are available from the corresponding authors upon request.

**Acknowledgments:** We thank M. Vicary from MRT for his contributions. We also thank J. Bowerman and C. Large, also from MRT, for the maps and cross-section publication. The authors published with the permission of the director of Mineral Resources Tasmania.

**Conflicts of Interest:** The authors declare no conflict of interest.

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
