# Peer review of "Insights and Lessons from 3D Geological and Geophysical Modeling of Mineralized Terranes in Tasmania"

_minerals, doi:10.3390/min11111195_

Round 1
Reviewer 1 Report
It is good to see that researchers and the exploration industry remain enthusiastic and innovative with 3D geological and geophysical modelling.
Minor comments included in the pdf.

Author Response
We thank the reviewer for constructive comments. We have corrected the manuscript with all the reviewers requests.
Reviewer 2 Report
Good and interesting paper. I have no any comments for inprovement.
Author Response
We thank the reviewer for their comments.
Reviewer 3 Report
The paper is well written and the topic is interesting for the scientific community; it is very detailed but, from my point of view, too long and dispersive, and so outcomes and improvements are taken to a second place.
It is not understandable if the work is aimed at the deposit description of the study area, or if it wants to provide the methodological indications for the realization of a 3D model.
The two aspects can be treated treatedly, however, guiding the reader more precisely on the method used to achieve the result.
It is therefore considered necessary to reorganize the paragraphs by dividing and schematizing the steps necessary for processing and the input data.
In this way the reader will be able to better understand what data is needed to obtain a result.
In the conclusions, it would also be useful to know the contribution of the individual parameters to the final result.
It is believed that a flowchart illustrating the steps and relationships in the process would be of great help to understanding.
Line 43, 147, 149, 164, 478: please write in the text the meaning of the abbreviations.
Line 363, figure 6: “ that conTable 3…” please check the typing error.
Line 443: Capitol letter after the full stop.
Line 782 ?
Figures:
Figure 9: please enlarge the size of the numbers of the scale Z, and the color of the axis X and Y, that are not visible
Figure 11: please enlarge the size of the numbers of the X- Y axis
Figure 18: improve the resolution of the image, especially the legend writings.
Figure 25: enlarge the size of x-y axis numbers
Author Response
We thank the reviewer for his constructive comments.
While we acknowledge the manuscript is long it is essentially a review paper of over two decades of research. However, we do announce new results of our modelling efforts in respective sections.
Tasmanian geology is some of the most complex on the planet and we’ve essentially divided it into two terranes (WTT, ETT). The 3D models correspond to these terranes. The reviewer’s suggestion of a flow chart was an excellent idea and we have included one to explain the 3D modelling methodology in section 3 and also added additional information of why we carry out our modelling this way.
We have also addressed all the reviewers suggested editing changes. For example the original The reviewers request for information on line 782 (now line 817) was the result of a missing (deleted) paragraph.
Regards
Daniel